

# GATI-RS model using Bi-LSTM and multi-head attention mechanism to enhance online shopping experience for the elderly with accurate click-through rate prediction

Ying Liu[1,2], Shahriman Zainal Abidin[1], Verly Veto Vermol[1], Shaolong Yang[2] and Hanyu Liu[2]

[1] Faculty of Art & Design, Universiti Teknologi MARA (UiTM), Selangor Darul Ehsan, Malaysia
[2] Zhengzhou Railway Vocational & Technical College, Zhengzhou, China

Corresponding author
Shahriman Zainal Abidin,
shahriman.z.a@uitm.edu.my

## ABSTRACT

With the rapid development of e-commerce and the increasing aging population, more elderly people are engaging in online shopping. However, challenges they face during this process are becoming more apparent. This article proposes a recommendation system based on click-through rate (CTR) prediction, aiming to enhance the online shopping experience for elderly users. By analyzing user characteristics, product features, and their interactions, we constructed a model combining bidirectional long short-term memory (Bi-LSTM) and multi-head self-attention mechanism to predict the item click behavior of elderly users in the recommendation section. Experimental results demonstrated that the model excels in CTR prediction, effectively improving the relevance of recommended content. Compared to the baseline model long short-term memory (LSTM), the GATI-RS framework improved CTR prediction accuracy by 40%, and its loss function rapidly decreased and remained stable during training. Additionally, the GATI-RS framework showed significant performance improvement when considering only elderly users, with accuracy surpassing the baseline model by 42%. These results indicate that the GATI-RS framework, through optimized algorithms, significantly enhances the model's global information integration and complex pattern recognition capabilities, providing strong support for developing recommendation systems for elderly online shoppers. This research not only offers new insights for e-commerce platforms to optimize services but also contributes to improving the quality of life and well-being of the elderly.

# INTRODUCTION
## Background

With the rapid development of e-commerce and the advent of an aging society, more elderly people are starting to shop online. However, they face unique challenges and needs when using e-commerce platforms, as shown in the figure. These challenges include difficulty adapting to complex interfaces, unfamiliarity with new technologies, and decreased ability to recognize information (*Gaiha, Lempert & Halpern-Felsher, 2020*;

*Smith et al., 2020*; *Abbott et al., 2023*). According to *Lawrence (2020)*, 75% of adults over 75 use the internet daily, with elderly people now accounting for 28% of all e-commerce purchases. Notably, women over 75 are among the fastest-growing groups of internet users. As the elderly become more familiar with technology, their online activities are increasing (*Lawrence, 2020*). Capital One Shopping reports that 41% of Baby Boomers and 28% of seniors prefer to purchase products and services online (*Capital One Shopping, 2025*). The increase in elderly users has highlighted the challenges they face in online shopping, including complex dynamic pricing, fake reviews, and unfamiliarity with technology use (*Bondi et al., 2021*). Elderly people need simpler forms, larger fonts, and more intuitive user interfaces to improve their online shopping experience (*Adchitects, 2025*).

Despite efforts by e-commerce platforms to simplify interfaces and functions, they still struggle to fully meet the needs of the elderly (*Chintala, Liaukonyte & Yang, 2024*; *Knutha & Ahrholdtb, 2022*). Elderly people are highly sensitive to discounts and promotions but often lack the technical means to fully utilize these offers, impacting their online shopping experience and satisfaction (*Smith et al., 2020*; *Son & Park, 2023*). In recent years, researchers and businesses have started focusing on how data analysis and machine learning can enhance the online shopping experience and engagement for the elderly. However, most existing click-through rate (CTR) prediction models have not fully incorporated age-related features or the unique behavioral patterns of elderly users, leading to less effective recommendations and engagement for this demographic. For instance, many CTR prediction models primarily focus on younger, tech-savvy user groups, neglecting the specific needs of elderly users (*Abbott et al., 2023*; *Smith et al., 2020*).

By analyzing elderly users' online shopping behavior and needs, we can provide more personalized and convenient shopping experiences, thereby improving their online shopping satisfaction and loyalty. Specific details are shown in Fig. 1.

In summary, predicting online shopping click behavior has become an important research direction. By building accurate CTR prediction models, e-commerce platforms can optimize recommendation systems, enhancing the relevance of recommended content and significantly improving the online shopping experience for elderly users. This study proposes a recommendation system (GATI-RS) combining bidirectional long short-term memory (Bi-LSTM) and multi-head self-attention mechanism, focusing on the unique needs of elderly users. Bi-LSTM effectively captures the temporal dependencies of user behavior, better recognizing user dynamics. The multi-head self-attention mechanism excels in handling global feature dependencies, providing more comprehensive feature integration, enabling the model to better serve elderly users. This research not only offers new methods for optimizing e-commerce recommendation systems but also helps elderly users find desired products more efficiently, enhancing their shopping experience.

## Literature review

With the rapid development of artificial intelligence (AI) and machine learning (ML), their applications in various fields are becoming increasingly widespread. In analyzing

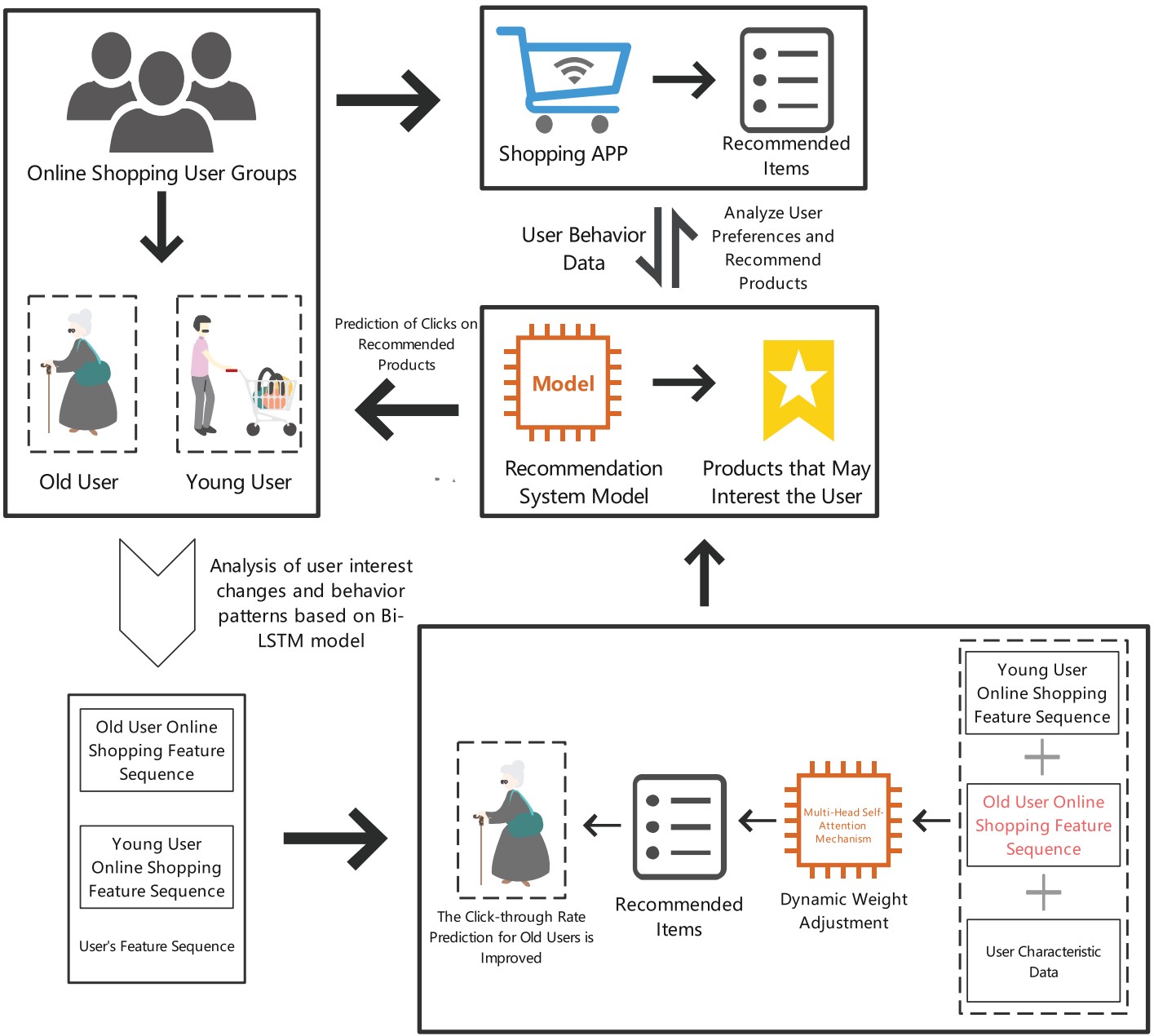

**Figure 1** Prediction results of item clicks in the recommendation section for elderly online shopping users.

existing recommendation systems, several key challenges emerge: handling temporal dependencies, addressing elderly user behavior patterns, and balancing model complexity with interpretability. Recent studies have explored various approaches but with notable limitations. *Rausch, Derra & Wolf (2022)* analyzed clickstream data and proposed

different ML methods to predict online shopping cart abandonment rates. While their methods achieved reasonable accuracy, they overlooked crucial temporal dependencies in user behavior and did not account for elderly users' unique shopping patterns, which often involve longer decision-making processes and different browsing behaviors. In the context of visual-based recommendations, *Li, Li & Shao (2016)* explored computer vision and pattern recognition applications in online recommendation systems, particularly in the fashion industry. Their approach demonstrated strong performance in image-based recommendations but faced challenges in capturing temporal dependencies and elderly user shopping behavior, which often requires more than visual cues alone. Addressing physical retail contexts, *Alfian et al. (2023)* employed radio frequency identification (RFID) technology and ML models to analyze in-store customer browsing activities. While their approach effectively captured physical browsing patterns, it did not translate well to online shopping scenarios where temporal characteristics play a crucial role. Similarly, *Beltzung et al. (2020)* focused on detecting fake stores using ML based on source code structure similarity, but their work did not address the temporal aspects of online shopping behavior. *Wang et al. (2022)* and *Zhou et al. (2022)* made significant contributions to online and manufacturing scheduling optimization. However, their methods' computational complexity limited their applicability to real-time recommendation scenarios, particularly for elderly users who may require more immediate responses.

Recent advances in preference modeling include *Kim & Zhang*'s *(2023)* attribute-based mixed-membership model for consumer preference structures. While innovative, their focus on the fashion market limits its generalizability, particularly for elderly users who may have different preference patterns across various product categories. *Wei et al. (2023)* proposed a dynamic Bayesian networks (DBN) based recommendation method that showed promising results. However, their model's complex architecture and high computational costs pose challenges for real-world deployment, especially in scenarios requiring quick responses for elderly users who may be less tolerant of system delays. Studies focusing on user behavior include *Beyari & Garamoun*'s *(2022)* investigation of AI tools' impact on online purchasing decisions. Their findings, while valuable, were limited to Saudi Arabian consumers and lacked validation across different cultural and age demographics, particularly the elderly population. *Damaševičius & Zailskaitė-Jakštė (2022)* proposed a Human-Centric Cyber Security model incorporating deep learning methods for click-through rate prediction, which achieved superior performance (AUC of 0.8062) on the Avazu dataset. While their work provides valuable insights into user behavior prediction and security, it primarily focuses on general online behavior patterns without specific consideration for age-specific user groups. In the domain of review analysis, *Bilal & Almazroi (2023)* and *Olmedilla, Rocio Martinez-Torres & Toral (2022)* proposed advanced methods using BERT and CNN respectively for review usefulness classification. While achieving high accuracy, these approaches primarily focused on general user populations without considering the specific needs and behaviors of elderly users.

**Table 1 Literature review summary.**

| Authors | Application scenario | Research content | Possible shortcomings |
|---|---|---|---|
| *Rausch, Derra & Wolf (2022)* | Online shopping cart abandonment prediction | Analyzed clickstream data, proposed different ML methods to predict online shopping cart abandonment rates. | Gradient boosting showed excellent performance, but it had heavy computational burden and ignored temporal dependencies and comprehensive behavior patterns of elderly users. |
| *Li, Li & Shao (2016)* | Enhancing online customer shopping experience | Explored applications of computer vision and pattern recognition in online recommendation systems, especially in the fashion industry. | Did not deeply investigate temporal dependencies and elderly user shopping behavior. |
| *Alfian et al. (2023)* | In-store customer shopping behavior analysis | Used RFID technology and ML models to analyze in-store customer browsing activities. | Focused on physical store contexts, did not consider temporal characteristics and overall dependencies in online shopping. |
| *Beltzung et al. (2020)* | Fake store detection | Used ML to detect fake stores, based on source code structure similarity for classification. | Focused on fake store detection, did not address temporal dependencies in online shopping. |
| *Wang et al. (2022)* | Online two-stage hybrid flow shop scheduling | Used independent double deep Q-network multi-agent reinforcement learning for online scheduling. | Complex RL model, did not effectively address temporal dependencies and elderly user behavior. |
| *Didden, Dang & Adan (2023)* | Online machine shop scheduling | Proposed decentralized multi-agent systems for online scheduling, where intelligent agents learn to respond autonomously to the environment. | Mainly focused on shop scheduling, did not consider temporal characteristics and overall behavior in online shopping. |
| *Zhou et al. (2022)* | Job shop scheduling optimization in intelligent manufacturing | Proposed intelligent schedulers using deep RL to optimize scheduling strategies, handling real-time jobs and emergencies. | Mainly applied to manufacturing, lacking research on temporal characteristics and overall dependencies in online shopping. |

More and more researchers are beginning to focus on these challenges and explore innovative solutions. However, there are still many shortcomings in previous studies, as shown in Table 1.

## Our contributions

- Proposed a model combining Bi-LSTM and multi-head self-attention mechanism to analyze user characteristics, product features, and their interactions, accurately predicting item click behavior of elderly users in the recommendation section. The Bi-LSTM effectively captures the temporal dependencies in user behavior, particularly for elderly users, who exhibit different shopping patterns. The multi-head self-attention mechanism addresses global feature dependencies, allowing the model to integrate various factors.

- Introduced a multi-head self-attention mechanism to handle global feature dependencies, enhancing the recommendation system's global information integration and complex pattern recognition capabilities. This improvement is especially beneficial for elderly users, whose purchasing decisions are influenced by multiple factors, such as special offers, technology unfamiliarity, and cognitive overload. The mechanism allows the model to provide more relevant, personalized, and diverse recommendations,

significantly improving the relevance of item clicks for elderly users in the recommendation section.

- Through experimental validation on real-world datasets, our model improved CTR prediction accuracy by 40%. Experimental results showed that the GATI-RS framework performed excellently in elderly user recommendation scenarios, with the loss function rapidly decreasing and remaining stable during training. These results highlight the model's ability to better adapt to elderly users' unique behavior patterns and improve their overall shopping experience.

## OUR APPROACH

### Problem description

We frame item click prediction as a classification task, predicting whether users will click on recommended items based on their behavior. Each user's behavior is captured in a $d$-dimensional feature vector $\mathbf{x}_i$, which includes browsing patterns, purchase history, and search queries.

$$\mathbf{x}_i = [x_{i1}, x_{i2}, \ldots, x_{id}] \quad \text{for} \quad i = 1, 2, \ldots, N \tag{1}$$

where $N$ represents the total user count, and $x_{ij}$ is the $j$-th feature for user $i$. We define the click behavior through a binary indicator $y_i$:

$$y_i = \begin{cases} 1 & \text{if user clicks the recommended item} \\ 0 & \text{otherwise.} \end{cases} \tag{2}$$

Using logistic regression, we estimate the click probability:

$$P(y_i = 1 | \mathbf{x}_i) = \frac{1}{1 + \exp(-\mathbf{w}^\top \mathbf{x}_i)} \tag{3}$$

with model parameters $\mathbf{w}$. The optimization uses log-loss:

$$L(\mathbf{w}) = -\frac{1}{N} \sum_{i=1}^{N} [y_i \log P(y_i = 1 | \mathbf{x}_i) + (1 - y_i) \log(1 - P(y_i = 1 | \mathbf{x}_i))]. \tag{4}$$

We optimize parameters through gradient descent:

$$\mathbf{w}^{(t+1)} = \mathbf{w}^{(t)} - \eta \nabla L(\mathbf{w}^{(t)}) \tag{5}$$

where $\eta$ denotes the learning rate.

**Problem 1** *Our core objective is to find optimal parameters $\mathbf{w}^*$ that maximize prediction accuracy:*

$$\mathbf{w}^* = \arg\max_{\mathbf{w}} \frac{1}{N} \sum_{i=1}^{N} [\delta(y_i, \mathbb{I}(P(y_i = 1 | \mathbf{x}_i) > 0.5))] \tag{6}$$

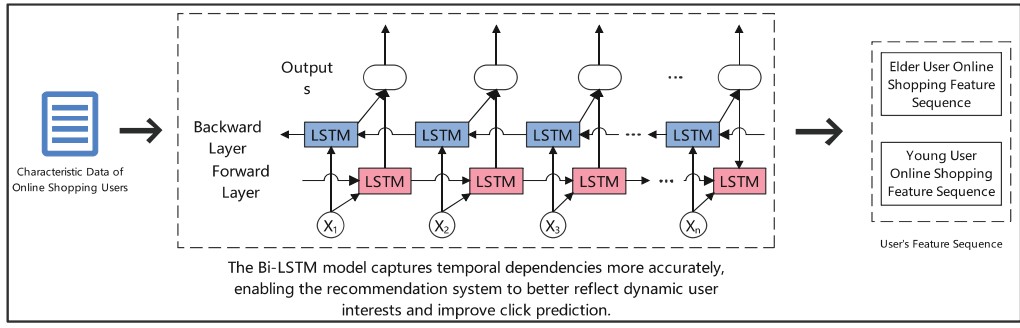

**Figure 2** Click prediction for elderly users in online shopping: analyzing behavior patterns with Bi-LSTM.

where $\mathbb{I}(\cdot)$ indicates the indicator function, and $\delta(a, b)$ represents the Kronecker delta function, evaluating to 1 when $a = b$ and 0 otherwise.

## Bi-LSTM: understanding user interest changes and behavior patterns

*Capturing temporal dependencies: user interest and behavior analysis with Bi-LSTM model*

- Current recommendation models face challenges in accurately tracking user behavior patterns, particularly in identifying dynamic interest shifts (*Wang et al., 2023*). These limitations result in decreased prediction accuracy and increased false recommendations (*Chao et al., 2023*; *Ni et al., 2023*).
- By implementing bidirectional data processing, our Bi-LSTM model effectively captures temporal patterns in user behavior sequences, enabling precise detection of interest evolution over time.

Figure 2 depicts the architecture of our Bi-LSTM model for click prediction. The model processes online shopping user characteristics through two parallel LSTM layers: a forward layer (shown in pink) and a backward layer (shown in blue). The forward layer analyzes the sequential progression of user behaviors, while the backward layer captures temporal dependencies in reverse order. Critical temporal features (marked as X1 to Xn) are processed bidirectionally to capture complex behavioral patterns. This dual-directional analysis is particularly effective for distinguishing between elderly and young user shopping patterns, as illustrated by the separate feature sequences on the right. The bidirectional structure enables more accurate temporal dependency modeling, ultimately enhancing the recommendation system's ability to predict click behaviors across different user demographics.

*Capturing temporal dependencies with Bi-LSTM model*

We employ the Bi-LSTM model to analyze temporal patterns in user behavior sequences. The user behavior at time step $t$ is represented as:

$$\mathbf{x}_t = [x_{t1}, x_{t2}, \ldots, x_{td}] \quad \text{for} \quad t = 1, 2, \ldots, T \tag{7}$$

where $d$ is the feature dimension and $T$ denotes time steps. In the forward LSTM, where $\mathbf{W}$ represents weight matrices, $\mathbf{b}$ represents bias vectors, and $\sigma$ is the sigmoid function:

$$\overrightarrow{\mathbf{i}}_t = \sigma\left(\mathbf{W}_{xi}\mathbf{x}_t + \mathbf{W}_{hi}\overrightarrow{\mathbf{h}}_{t-1} + \mathbf{b}_i\right)$$
$$\overrightarrow{\mathbf{f}}_t = \sigma\left(\mathbf{W}_{xf}\mathbf{x}_t + \mathbf{W}_{hf}\overrightarrow{\mathbf{h}}_{t-1} + \mathbf{b}_f\right)$$
$$\overrightarrow{\mathbf{c}}_t = \overrightarrow{\mathbf{f}}_t \odot \overrightarrow{\mathbf{c}}_{t-1} + \overrightarrow{\mathbf{i}}_t \odot \tanh\left(\mathbf{W}_{xc}\mathbf{x}_t + \mathbf{W}_{hc}\overrightarrow{\mathbf{h}}_{t-1} + \mathbf{b}_c\right) \quad (8)$$
$$\overrightarrow{\mathbf{o}}_t = \sigma\left(\mathbf{W}_{xo}\mathbf{x}_t + \mathbf{W}_{ho}\overrightarrow{\mathbf{h}}_{t-1} + \mathbf{b}_o\right)$$
$$\overrightarrow{\mathbf{h}}_t = \overrightarrow{\mathbf{o}}_t \odot \tanh(\overrightarrow{\mathbf{c}}_t).$$

Similarly for backward LSTM, with $\overleftarrow{\mathbf{h}}_t$ and $\overleftarrow{\mathbf{c}}_t$ representing hidden and cell states:

$$\overleftarrow{\mathbf{i}}_t = \sigma\left(\mathbf{W}_{xi}\mathbf{x}_t + \mathbf{W}_{hi}\overleftarrow{\mathbf{h}}_{t+1} + \mathbf{b}_i\right)$$
$$\overleftarrow{\mathbf{f}}_t = \sigma\left(\mathbf{W}_{xf}\mathbf{x}_t + \mathbf{W}_{hf}\overleftarrow{\mathbf{h}}_{t+1} + \mathbf{b}_f\right)$$
$$\overleftarrow{\mathbf{c}}_t = \overleftarrow{\mathbf{f}}_t \odot \overleftarrow{\mathbf{c}}_{t+1} + \overleftarrow{\mathbf{i}}_t \odot \tanh\left(\mathbf{W}_{xc}\mathbf{x}_t + \mathbf{W}_{hc}\overleftarrow{\mathbf{h}}_{t+1} + \mathbf{b}_c\right) \quad (9)$$
$$\overleftarrow{\mathbf{o}}_t = \sigma\left(\mathbf{W}_{xo}\mathbf{x}_t + \mathbf{W}_{ho}\overleftarrow{\mathbf{h}}_{t+1} + \mathbf{b}_o\right)$$
$$\overleftarrow{\mathbf{h}}_t = \overleftarrow{\mathbf{o}}_t \odot \tanh(\overleftarrow{\mathbf{c}}_t).$$

The Bi-LSTM output combines forward and backward hidden states:

$$\mathbf{h}_t = \left[\overrightarrow{\mathbf{h}}_t; \overleftarrow{\mathbf{h}}_t\right]. \quad (10)$$

The final prediction uses weight matrix $\mathbf{W}_o$ and bias vector $\mathbf{b}_o$:

$$\mathbf{o}_t = \mathbf{W}_o\left(\overrightarrow{\mathbf{h}}_t \oplus \overleftarrow{\mathbf{h}}_t\right) + \mathbf{b}_o = \mathbf{W}_o\left(\begin{bmatrix}\overrightarrow{\mathbf{h}}_t \\ \overleftarrow{\mathbf{h}}_t\end{bmatrix}\right) + \mathbf{b}_o. \quad (11)$$

The loss function with weight $w_t$ at time step $t$ is defined as:

$$L = -\frac{1}{T}\sum_{t=1}^{T}\left[w_t y_t \log P(y_t = 1|\mathbf{h}_t) + (1 - w_t)(1 - y_t)\log P(y_t = 0|\mathbf{h}_t)\right]. \quad (12)$$

The click and non-click probabilities are calculated as:

$$P(y_t = 1|\mathbf{h}_t) = \frac{1}{1 + \exp\left(-\left(\mathbf{W}_o\left(\overrightarrow{\mathbf{h}}_t \oplus \overleftarrow{\mathbf{h}}_t\right) + \mathbf{b}_o\right)\right)} \quad (13)$$

$$P(y_t = 0|\mathbf{h}_t) = 1 - \frac{1}{1 + \exp\left(-\left(\mathbf{W}_o\left(\overrightarrow{\mathbf{h}}_t \oplus \overleftarrow{\mathbf{h}}_t\right) + \mathbf{b}_o\right)\right)}. \quad (14)$$

The optimization objective for parameters $\mathbf{W}_o$ and $\mathbf{b}_o$ is:

$$(\mathbf{W}_o, \mathbf{b}_o)^* = \arg\min_{\mathbf{W}_o, \mathbf{b}_o} \left\{ -\frac{1}{T} \sum_{t=1}^{T} \left[ w_t y_t \log\left( \frac{1}{1 + \exp\left(-\left(\mathbf{W}_o\left(\overrightarrow{\mathbf{h}_t} \oplus \overleftarrow{\mathbf{h}_t}\right) + \mathbf{b}_o\right)\right)} \right) \right. \right. \tag{15}$$
$$\left. \left. + (1 - w_t)(1 - y_t) \log\left( 1 - \frac{1}{1 + \exp\left(-\left(\mathbf{W}_o\left(\overrightarrow{\mathbf{h}_t} \oplus \overleftarrow{\mathbf{h}_t}\right) + \mathbf{b}_o\right)\right)} \right) \right] \right\}.$$

**Theorem 1** *The Bi-LSTM model converges to a unique solution* $(\mathbf{W}_o, \mathbf{b}_o)^*$, *where* $\sigma(\cdot)$ *is the sigmoid function,* $\lambda$ *is the regularization parameter, and* $||\mathbf{W}_o||_F^2$ *is the Frobenius norm:*

$$\mathcal{L}(\mathbf{W}_o, \mathbf{b}_o, \mathbf{H}) = \frac{1}{T} \sum_{t=1}^{T} \left[ y_t \log \sigma\left(\mathbf{W}_o\left(\overrightarrow{\mathbf{h}_t} \oplus \overleftarrow{\mathbf{h}_t}\right) + \mathbf{b}_o\right) \right. \tag{16}$$
$$\left. + (1 - y_t) \log\left(1 - \sigma\left(\mathbf{W}_o\left(\overrightarrow{\mathbf{h}_t} \oplus \overleftarrow{\mathbf{h}_t}\right) + \mathbf{b}_o\right)\right) \right] + \lambda ||\mathbf{W}_o||_F^2.$$

**Corollary 1** *The optimized parameters enhance attention mechanism performance, where* $\mathbf{W}_Q, \mathbf{W}_K, \mathbf{W}_V$ *are learnable parameter matrices:*

$$\mathbf{Q}, \mathbf{K}, \mathbf{V} = \mathbf{H}\mathbf{W}_Q, \mathbf{H}\mathbf{W}_K, \mathbf{H}\mathbf{W}_V \quad where \quad \mathbf{W}_Q, \mathbf{W}_K, \mathbf{W}_V \in \mathbb{R}^{d \times d}. \tag{17}$$

*The attention weights are computed as follows, where* $d_k$ *is the dimension of the key vector:*

$$Attention(\mathbf{Q}, \mathbf{K}, \mathbf{V}) = \mathbf{H} \cdot softmax\left( \frac{\mathbf{W}_Q \mathbf{W}_K^\top \mathbf{H}^\top \mathbf{H}}{\sqrt{d_k}} \right) \mathbf{W}_V. \tag{18}$$

For detailed evidence, see the Appendix "Mathematical Theorems and Corollary Proofs".

## Multi-head self-attention mechanism
### *Capturing global dependencies: multimodal feature fusion and analysis with multi-head self-attention mechanism*

- Static feature analysis in conventional models overlooks crucial interactions between different feature types (*Salachoris et al., 2024*), leading to accuracy degradation and increased maintenance overhead (*Van Thieu & Mirjalili, 2023*; *Rahman et al., 2023*).
- The multi-head self-attention mechanism enables parallel processing of multimodal features, facilitating dynamic importance adjustment and comprehensive feature fusion. This approach particularly enhances recommendation precision for elderly users by capturing their unique shopping patterns.

Figure 3 illustrates the architecture of our multi-head attention mechanism for personalized recommendation. The model integrates scaled dot-product attention with multi-head structure to capture global feature dependencies, specifically designed to
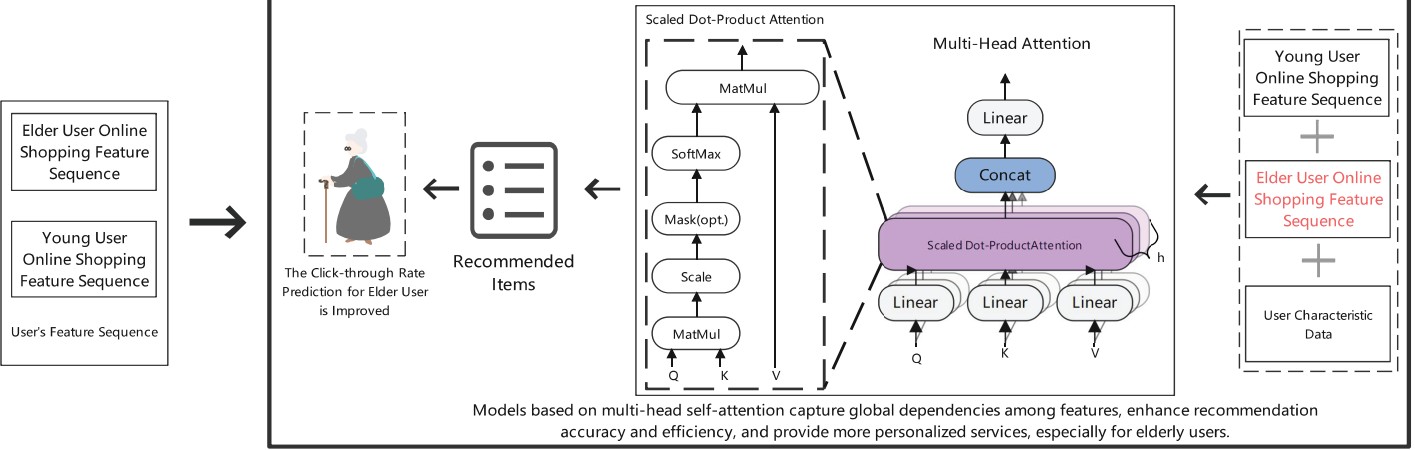

**Figure 3** Prediction results of item clicks in the recommendation section for elderly online shopping users.

enhance recommendation accuracy for elderly users by processing different user characteristic sequences in parallel.

### Capturing global dependencies with multi-head self-attention mechanism

For elderly-oriented shopping recommendations, we employ multi-head self-attention to analyze multimodal user behaviors. The mechanism integrates three key components: user features $\mathbf{U}$, item features $\mathbf{P}$, and temporal behavior patterns $\mathbf{H} = \{\mathbf{h}_t\}_{t=1}^{T}$ from Bi-LSTM:

$$\mathbf{X} = \begin{bmatrix} \mathbf{U} \\ \mathbf{P} \\ \mathbf{H} \end{bmatrix} \tag{19}$$

where $\mathbf{U} \in \mathbb{R}^{d_u}$ captures user preferences, $\mathbf{P} \in \mathbb{R}^{d_p}$ represents item characteristics, $\mathbf{H} \in \mathbb{R}^{T \times d_h}$ encodes temporal patterns, and the combined feature matrix $\mathbf{X} \in \mathbb{R}^{(d_u + d_p + T \cdot d_h) \times d}$. The feature transformation for attention computation is:

$$\mathbf{Q} = \mathbf{X}\mathbf{W}_Q, \quad \mathbf{K} = \mathbf{X}\mathbf{W}_K, \quad \mathbf{V} = \mathbf{X}\mathbf{W}_V \tag{20}$$

where $\mathbf{W}_Q, \mathbf{W}_K, \mathbf{W}_V \in \mathbb{R}^{d \times d}$ are learnable weight matrices for query, key, and value transformations respectively, adapting to elderly shopping patterns. The attention mechanism computes behavioral relevance through:

$$\text{Attention}(\mathbf{Q}, \mathbf{K}, \mathbf{V}) = \text{softmax}\left(\frac{\mathbf{Q}\mathbf{K}^{\top}}{\sqrt{d_k}}\right)\mathbf{V} \tag{21}$$

where $d_k$ is the dimension of the key vector, and the scaling factor $\sqrt{d_k}$ prevents attention scores from becoming too large in magnitude. To capture diverse shopping preferences of elderly users, we employ multiple attention heads:

$$\text{MultiHead}(\mathbf{Q}, \mathbf{K}, \mathbf{V}) = \text{Concat}(\text{head}_1, \text{head}_2, \dots, \text{head}_h)\mathbf{W}_O \tag{22}$$

where $h$ is the number of attention heads and $\mathbf{W}_O$ is the output projection matrix. Each attention head processes specific behavioral aspects:

$$\text{head}_i = \text{Attention}(\mathbf{Q}_i, \mathbf{K}_i, \mathbf{V}_i) \text{ where } \mathbf{Q}_i = \mathbf{X}\mathbf{W}_Q^i, \mathbf{K}_i = \mathbf{X}\mathbf{W}_K^i, \mathbf{V}_i = \mathbf{X}\mathbf{W}_V^i \tag{23}$$

where $\mathbf{W}_Q^i, \mathbf{W}_K^i, \mathbf{W}_V^i \in \mathbb{R}^{d \times d_h}$ are the weight matrices for the $i$-th head. The integrated output combines all behavioral patterns:

$$\mathbf{O} = \text{Concat}(\text{head}_1, \text{head}_2, \dots, \text{head}_h)\mathbf{W}_O \tag{24}$$

where $\mathbf{W}_O \in \mathbb{R}^{(h \cdot d_h) \times d}$ is the output weight matrix. For optimizing elderly user recommendations, we define the loss function as:

$$
\begin{aligned}
L = -\frac{1}{N} \sum_{i=1}^{N} \Bigg[ & y_i \log\left( \frac{1}{1 + \exp(-(\mathbf{W}_O(\text{Concat}(\text{Attention}(\mathbf{Q}_1, \mathbf{K}_1, \mathbf{V}_1), \dots, \text{Attention}(\mathbf{Q}_h, \mathbf{K}_h, \mathbf{V}_h))))))} \right) \\
& + (1 - y_i) \log\left( 1 - \frac{1}{1 + \exp(-(\mathbf{W}_O(\text{Concat}(\text{Attention}(\mathbf{Q}_1, \mathbf{K}_1, \mathbf{V}_1), \dots, \text{Attention}(\mathbf{Q}_h, \mathbf{K}_h, \mathbf{V}_h))))))} \right) \Bigg]
\end{aligned}
\tag{25}
$$

where $N$ is the number of samples and $y_i$ represents the actual click behavior. The click probabilities for recommendation items are computed as:

$$P(y_i = 1 | \mathbf{O}_i) = \frac{1}{1 + \exp(-\mathbf{O}_i)} \tag{26}$$

where $\mathbf{O}_i$ is the model output for the $i$-th sample. The optimal parameters are found through:

$$\theta^* = \arg\min_{\theta} L(\theta) \tag{27}$$

where $\theta$ represents all model parameters, including $\mathbf{W}_Q, \mathbf{W}_K, \mathbf{W}_V$, and $\mathbf{W}_O$.

**Theorem 2** *For elderly-oriented recommendations, the model converges to optimal parameters $\theta^*$ that minimize:*

$$
\begin{aligned}
\mathcal{L}(\theta) = \frac{1}{N} \sum_{i=1}^{N} & [y_i \log \sigma(O_i) + (1 - y_i) \log(1 - \sigma(O_i))] \\
& + \lambda \sum_{i=1}^{h} \left( \|W_Q^i\|_F^2 + \|W_K^i\|_F^2 + \|W_V^i\|_F^2 \right)
\end{aligned}
\tag{28}
$$

*where $\sigma(\cdot)$ denotes the sigmoid function, $\lambda$ is the regularization parameter, and $||\cdot||_F^2$ is the Frobenius norm.*

**Corollary 2.** *The attention weights adapt dynamically to elderly users' behavior patterns through:* [

$$
\mathbf{A} = softmax\left( \frac{1}{\sqrt{d_k}} \left( \begin{bmatrix} \mathbf{U}_1, \mathbf{U}_2, \dots, \mathbf{U}_{d_u} \\ \mathbf{P}_1, \mathbf{P}_2, \dots, \mathbf{P}_{d_p} \\ \mathbf{H}_{11}, \mathbf{H}_{12}, \dots, \mathbf{H}_{1d_h} \\ \mathbf{H}_{21}, \mathbf{H}_{22}, \dots, \mathbf{H}_{2d_h} \\ \vdots \\ \mathbf{H}_{T1}, \mathbf{H}_{T2}, \dots, \mathbf{H}_{Td_h} \end{bmatrix} \begin{bmatrix} \mathbf{w}_{Q1}, \mathbf{w}_{Q2}, \dots, \mathbf{w}_{Qd} \end{bmatrix} \right) \left( \begin{bmatrix} \mathbf{U}_1, \mathbf{U}_2, \dots, \mathbf{U}_{d_u} \\ \mathbf{P}_1, \mathbf{P}_2, \dots, \mathbf{P}_{d_p} \\ \mathbf{H}_{11}, \mathbf{H}_{12}, \dots, \mathbf{H}_{1d_h} \\ \mathbf{H}_{21}, \mathbf{H}_{22}, \dots, \mathbf{H}_{2d_h} \\ \vdots \\ \mathbf{H}_{T1}, \mathbf{H}_{T2}, \dots, \mathbf{H}_{Td_h} \end{bmatrix} \begin{bmatrix} \mathbf{w}_{K1}, \mathbf{w}_{K2}, \dots, \mathbf{w}_{Kd} \end{bmatrix} \right)^{\top} \right) \tag{29}
$$

] *where* **A** *represents the attention weight matrix that captures feature interactions, with components* $\mathbf{U}_i$, $\mathbf{P}_i$, *and* $\mathbf{H}_{ij}$ *representing user features, product features, and temporal behavior features respectively.*

For detailed evidence, see the Appendix "Mathematical Theorems and Corollary Proofs".

## GATI-RS model

The space complexity of the algorithm is mainly: the input feature vector dimension is $d$, and the hidden layer dimension is $h$. The total number of parameters for each LSTM is $4(dh + hh + h)$. Therefore, the total number of parameters for Bi-LSTM is $2 \times 4(dh + hh + h)$. The attention mechanism parameters include the query, key, and value weight matrices, each with dimensions $d \times d$. There are $h$ heads in the multi-head attention mechanism, each with $3 \times d \times d$ weight matrix parameters. Thus, the total number of parameters is $h \times 3 \times d \times d$. The number of parameters in the output layer depends on the dimension of the output vector and the hidden state. Assuming the dimension of the output vector is $o$, the number of parameters in the output layer is $hd \times o$. In summary, the algorithm's space complexity is $O(8dh^2 + 3hd^2 + hdo)$. The time complexity of the algorithm is mainly: the computation complexity for each time step is $O(dh + h^2)$, and the total number of time steps is $T$. Therefore, the time complexity of Bi-LSTM is $O(T(dh + h^2))$. The attention mechanism computes the query, key, and value matrices, as well as the attention weights and output. The computation complexity for the query, key, and value matrices is $O(d^2)$, the attention weight computation complexity is $O(d^2 + d)$, and the attention output computation complexity is $O(d^2)$. The total time complexity is $O(h \times (d^2 + d^2 + d)) = O(hd^2)$. The computation complexity of the output layer is $O(hdo)$. In summary, the algorithm's time complexity is $O(T(dh + h^2) + hd^2 + hdo)$.

As shown in Table 2, the time and space complexities of different models vary significantly, each with its own strengths and trade-offs. The proposed GATI-RS model demonstrates a relatively balanced approach, with time complexity $O(T(dh + h^2) + hd^2 + hdo)$, which is well-suited for large-scale and long-sequence data processing. However, this complexity might raise concerns for very large datasets in certain scenarios. In comparison, the Complex Network Model's time complexity of $O(N \cdot \log(N))$ is relatively more efficient in terms of computation for large-scale data but could lead to performance issues when data becomes too extensive.

Regarding space complexity, the GATI-RS model, with a space complexity of $O(8dh^2 + 3hd^2 + hdo)$, is optimized for handling multi-head self-attention mechanisms efficiently, which are crucial in recommendation tasks involving high-dimensional data. In contrast, the Complex Network Model's space complexity of $O(N^2)$ may cause memory bottlenecks when dealing with large-scale network data. While models like DURation, DiffRec, and NSRM excel in specific scenarios, they do not offer the same balance of time and space complexity as GATI-RS. Overall, the GATI-RS model offers a more adaptable and efficient solution when considering both time and space complexities for complex recommendation tasks.

---

**Algorithm 1  Pseudo-code for item click prediction model.**

Input: User features $\mathbf{U}$, item features $\mathbf{P}$, user behavior sequence $\{\mathbf{x}_t\}_{t=1}^{T}$

Initialize: Bi-LSTM parameters $\mathbf{W}_{xi}, \mathbf{W}_{hi}, \mathbf{b}_i, \mathbf{W}_{xf}, \mathbf{W}_{hf}, \mathbf{b}_f, \mathbf{W}_{xc}, \mathbf{W}_{hc}, \mathbf{b}_c, \mathbf{W}_{xo}, \mathbf{W}_{ho}, \mathbf{b}_o$

**for** each time step $t$ from 1 to $T$ **do**

    Compute Bi-LSTM forward hidden state $\overrightarrow{\mathbf{h}}_t$ and cell state $\overrightarrow{\mathbf{c}}_t$ (Eq. (8))

    Compute Bi-LSTM backward hidden state $\overleftarrow{\mathbf{h}}_t$ and cell state $\overleftarrow{\mathbf{c}}_t$ (Eq. (9))

**end for**

Concatenate forward and backward hidden states $\mathbf{h}_t = [\overrightarrow{\mathbf{h}}_t; \overleftarrow{\mathbf{h}}_t]$ (Eq. (10))

Compute output $\mathbf{o}_t = \mathbf{W}_o\left(\overrightarrow{\mathbf{h}}_t \oplus \overleftarrow{\mathbf{h}}_t\right) + \mathbf{b}_o$ (Eq. (11))

Compute loss function $L$ as weighted cross-entropy loss (Eq. (12))

$$L = -\frac{1}{T}\sum_{t=1}^{T}[w_t y_t \log P(y_t = 1|\mathbf{h}_t) + (1 - w_t)(1 - y_t)\log P(y_t = 0|\mathbf{h}_t)]$$

Minimize loss function using gradient descent, updating model parameters (Eq. (15))

Initialize multi-head self-attention mechanism: weight matrices $\mathbf{W}_Q, \mathbf{W}_K, \mathbf{W}_V, \mathbf{W}_O$

Compute query matrix $\mathbf{Q}$, key matrix $\mathbf{K}$, and value matrix $\mathbf{V}$ (Eq. (20))

$$\mathbf{Q} = \mathbf{X}\mathbf{W}_Q, \quad \mathbf{K} = \mathbf{X}\mathbf{W}_K, \quad \mathbf{V} = \mathbf{X}\mathbf{W}_V$$

Compute attention weights and output through self-attention mechanism (Eq. (21))

$$\text{Attention}(\mathbf{Q}, \mathbf{K}, \mathbf{V}) = \text{softmax}\left(\frac{\mathbf{Q}\mathbf{K}^{\top}}{\sqrt{d_k}}\right)\mathbf{V}$$

Concatenate multiple heads' outputs and transform through a linear layer (Eq. (24))

$$\mathbf{O} = \text{Concat}(\text{head}_1, \text{head}_2, \ldots, \text{head}_h)\mathbf{W}_O$$

Compute final click probability prediction (Eq. (26))

$$P(y_i = 1|\mathbf{O}_i) = \frac{1}{1 + \exp(-\mathbf{O}_i)}$$

Optimize model parameters to minimize the loss function (Eq. (27))

$$\theta^* = \arg\min_{\theta} L(\theta)$$

Output: Click probability prediction $\hat{y}$

---

**Table 2  Comparison of time and space complexity of recommendation system models.**

| Model name | Time complexity | Space complexity |
|---|---|---|
| GATI-RS | $O(T(dh + h^2) + hd^2 + hdo)$ | $O(8dh^2 + 3hd^2 + hdo)$ |
| DURation (Lu et al., 2022) | $O(N \cdot d \cdot k)$ | $O(d^2 + N \cdot d \cdot k)$ |
| Complex Network Model (Alhadlaq, Kerrache & Aboalsamh, 2022) | $O(N \cdot \log(N))$ | $O(N^2)$ |
| TSA-RM (Xingyao et al., 2022) | $O(T(dh + h^2) + hd^2)$ | $O(8dh^2 + 3hd^2 + hdo)$ |
| DiffRec (Du et al., 2023) | $O(T \cdot (d^2 + d))$ | $O(d^2 + d)$ |
| NSRM (Wu et al., 2023) | $O(N \cdot d \cdot \log(N))$ | $O(d^2 + N \cdot d \cdot k)$ |

# EXPERIMENTS AND RESULTS

## Dataset and experimental parameters

**Dataset description (https://zenodo.org/records/14198398):** In this study, we employ a comprehensive e-commerce dataset from Alibaba Cloud Tianchi platform, which is

**Table 3 Comprehensive dataset description.**

| Category | Feature | Type | Description |
|---|---|---|---|
| Dataset scale | Training users | 5,752,528 | Number of unique users in training set |
| | Training items | 31,431,181 | Number of unique items in training set |
| | Training samples | 9,526,571 | Total training interactions |
| | Test users | 2,242,720 | Number of unique users in test set |
| | Test items | 17,877,169 | Number of unique items in test set |
| User features | user | Scalar | Encrypted user ID identifier |
| | user_gender | Scalar | Encrypted gender ID |
| | user_age_level | Scalar | Encrypted age group indicator |
| | user_purchase_level | Scalar | Encrypted purchase power rating |
| | user_hour | Scalar | Encrypted interaction time |
| Item features | cand_item_pos | Scalar | Item position in display sequence |
| | cand_item_pagenum | Scalar | Page number placement |
| | cand_item_sex | Scalar | Gender preference indicator |
| | cand_item_price_level | Scalar | Item price level |
| | cand_item_age_level | Scalar | Target age level |
| | cand_item_bc_type | Scalar | Business category type |
| Exposure sequence | exp_item_pos_seq | Sequence | Exposure position sequence (padded to 50) |
| | exp_item_pagenum_seq | Sequence | Page number sequence |
| | exp_item_sex_seq | Sequence | Gender preference sequence |
| | exp_item_price_level_seq | Sequence | Price level sequence |
| | exp_item_age_level_seq | Sequence | Age level sequence |
| | exp_item_bc_type_seq | Sequence | Category type sequence |
| Click sequence | ipv_item_pos_seq | Sequence | Click position sequence (padded to 30) |
| | ipv_item_pagenum_seq | Sequence | Click page number sequence |
| | ipv_item_sex_seq | Sequence | Click gender preference sequence |
| | ipv_item_price_level_seq | Sequence | Click price level sequence |
| | ipv_item_age_level_seq | Sequence | Click age level sequence |
| | ipv_item_bc_type_seq | Sequence | Click category type sequence |
| Labels | Label | Scalar | Binary indicator for click events |

publicly available under the CC-BY-SA-NC 4.0 license. The dataset exhibits significant scale, as shown in Table 3, containing 9,526,571 interaction samples from 5,752,528 unique users and 31,431,181 items in the training set, and 3,555,419 samples from 2,242,720 users and 17,877,169 items in the test set. The data captures user interactions through a paginated interface where multiple items are displayed simultaneously, recording both exposure and click events in each session. The dataset features encrypted user-item interaction logs, including user demographics, item characteristics, and behavioral sequences. Notably, the item positions in the dataset are non-continuous due to the interspersion of other business content, reflecting authentic e-commerce scenarios.

**Computing infrastructure**: The experiments were conducted on a system running Windows OS, with hardware configurations including an Intel Core i7-9700K CPU, 16 GB

RAM, and an NVIDIA GeForce RTX 2070 GPU. We used Python 3.8 and TensorFlow 2.x for development and experiments.

**Data preprocessing**: Prior to model training, we conducted comprehensive data preprocessing. Numerical features were standardized using z-score normalization to address varying scales and improve model convergence. Categorical variables underwent one-hot encoding to prevent artificial ordinal relationships. Missing values were handled using different strategies based on feature characteristics: median imputation for skewed numerical distributions, mean imputation for normal distributions, and mode imputation with missing indicators for categorical features. This hybrid approach was chosen to preserve the underlying patterns in elderly user behavior while maintaining data quality. Additionally, we engineered temporal features by extracting cyclical components and creating time-window based aggregations to better capture usage patterns specific to elderly users.

The specific parameters used in the experiment are shown in Table 4. Through extensive experimentation and ablation studies, key hyperparameters were optimized: a learning rate of 0.001 was selected *via* grid search, and a batch size of 256 was chosen for optimal training efficiency. The model employs a three-layer neural network (128/256/128 neurons) with ReLU activation. To prevent overfitting, we implemented L2 regularization (0.0005), dropout (0.5), and early stopping after five epochs without improvement. The attention mechanism uses eight heads with dimension 64, while the Bi-LSTM component consists of two layers with 128 hidden units, effectively capturing temporal dependencies in the data.

The sample size of the test dataset in this study is 2,242,720 users, and 17,877,169 items, with a total of 9,526,571 interactions in the training set. To analyze the importance of users in different age groups in online shopping, users are divided into young, middle-aged, and elderly stages. As shown in Table 5, elderly users account for 23.34% of our dataset, which is close to one-fourth. Although middle-aged users represent the largest proportion at 60.73%, elderly users remain a significant group for online shopping. This group often demonstrates higher purchasing power and loyalty, especially in areas such as health products, household items, and healthcare products. Therefore, optimizing for the preferences and needs of elderly users can significantly enhance user satisfaction and sales on e-commerce platforms.

## Framework experimental results

To fully validate our model design and demonstrate the necessity of each component, we conducted comprehensive ablation studies. We compared three model variants: (1) the complete GATI-RS framework with multi-head self-attention, (2) GATI-RS without the attention mechanism, and (3) a baseline LSTM model. These experiments were conducted on both general users and elderly users to verify the effectiveness of our design choices.

### Click prediction for online shopping users

As shown in Fig. 4A, we conducted control experiments with different optimization schemes to analyze the model performance. Using LSTM as the baseline model enables

**Table 4 Detailed experimental parameter settings.**

| Parameter name | Parameter value | Parameter name | Parameter value |
|---|---|---|---|
| Dataset | Custom e-commerce dataset | Number of epochs | 50 |
| Number of training samples | 9,526,571 | Number of test samples | 3,555,419 |
| Number of neurons per layer | 128/256/128 | Learning rate | 0.001 |
| Batch size | 256 | Number of iterations | 20 |
| Optimizer | Adam | Activation function | ReLU |
| Regularization | L2 | Regularization parameter | 0.0005 |
| Early stopping criterion | 5 epochs without improvement | Data augmentation | None |
| Data preprocessing | Standardization | Loss function | Cross-entropy |
| Evaluation metrics | Accuracy (ACC), LOSS | Training/validation ratio | 80%/20% |
| Feature engineering | User and item feature embedding | Bi-LSTM parameters | Hidden units = 128, Layers = 2 |
| Self-attention mechanism parameters | Number of heads = 8, Dimension = 64 | Attention dropout rate | 0.1 |
| Number of hidden layers | 3 | Dropout rate | 0.5 |
| Output layer configuration | Fully connected, Sigmoid | Model saving | Best model |
| Computational resources | GPU Tesla V100 | Feature fusion strategy | Concatenation |
| Weight initialization | Xavier initialization | Number of time steps | 10 |
| Sequence length | 20 | Bi-LSTM hidden state dimension | 64 |

**Table 5 Proportion of users in different age groups.**

| Age group | Number of users | Proportion |
|---|---|---|
| Young | 567,133 | 15.93% |
| Middle-aged | 2,158,123 | 60.73% |
| Elderly | 830,163 | 23.34% |

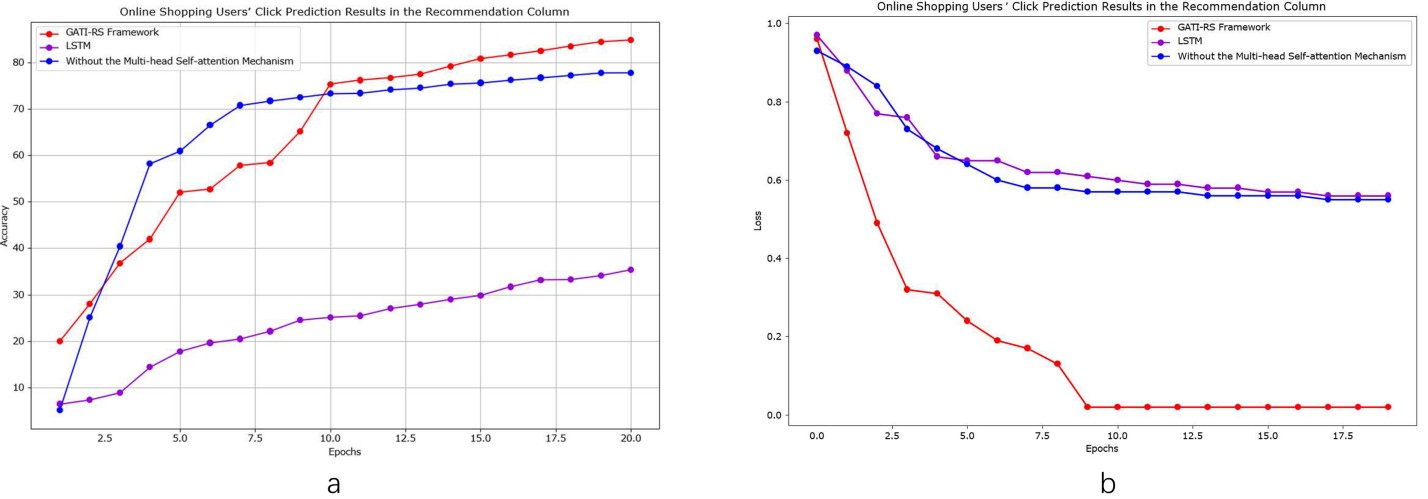

a

b

**Figure 4 Performance evaluation of GATI-RS framework against baseline models in online shopping recommendation tasks.**

objective evaluation and clearly demonstrates the improvements achieved by the GATI-RS framework under various optimization algorithms. The experimental results confirm that the GATI-RS framework achieves superior performance across different optimization configurations. As shown in Fig. 4, we conducted control experiments with different optimization schemes to analyze model performance. Using LSTM as the baseline enables clear demonstration of performance improvements achieved by the GATI-RS framework under various optimization algorithms.

In the experimental results (Fig. 4A), when excluding the multi-head self-attention mechanism, prediction accuracy decreases by approximately 10% compared to the complete GATI-RS framework. Using LSTM alone results in a 40% lower accuracy. While both optimization algorithms show improvements in click-through rate prediction for online shopping recommendations, the baseline LSTM model performs notably worse, with accuracy approximately 30% below the GATI-RS framework.

The loss function changes during training provide additional insights into model performance (Fig. 4B). The GATI-RS framework demonstrates superior optimization, with its loss function steadily decreasing and stabilizing. Models without multi-head self-attention show rapid initial loss decrease but stabilize at a higher value than GATI-RS, indicating limitations in capturing global dependencies. The LSTM baseline exhibits the slowest convergence and highest final loss, suggesting insufficient capability in complex pattern recognition and global information learning.

Figure 5 presents the Area Under the Curve (AUC) results for each model configuration. The GATI-RS framework, incorporating the multi-head self-attention mechanism, achieved the highest AUC score of approximately 0.92. This result demonstrates the model's strong ability to distinguish between positive and negative classes, indicating superior predictive power and feature integration capabilities. In contrast, the GATI-RS framework without the multi-head self-attention mechanism scored an AUC of approximately 0.82, highlighting the importance of the self-attention mechanism in capturing global feature dependencies. The LSTM baseline model showed the lowest performance with an AUC of around 0.62, underscoring its limited capacity to handle complex patterns and global information. These results confirm that the GATI-RS framework with the multi-head self-attention mechanism outperforms other configurations in the normal user group, providing more accurate and relevant recommendations.

### Click prediction for elderly online shopping users

As shown in Fig. 6, we conducted detailed comparative analyses focusing specifically on elderly online shopping users. From Fig. 6A, the model without multi-head self-attention achieves relatively high accuracy but remains approximately 7% below the GATI-RS framework. While this simplified model offers reduced computational complexity, its limited global information integration capability results in inferior pattern recognition performance. The baseline LSTM model shows the lowest accuracy, approximately 40% below GATI-RS, despite its theoretical advantages in temporal data processing. This

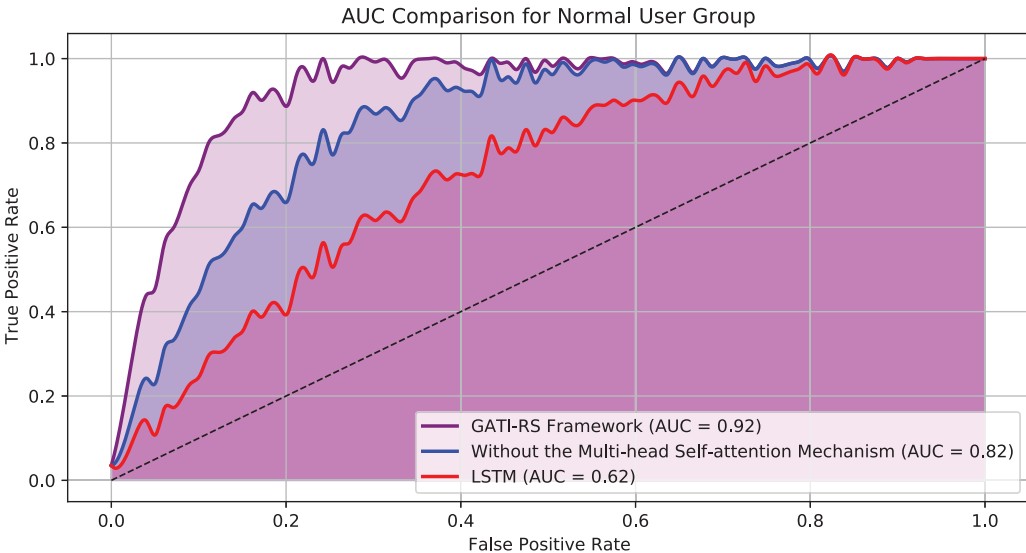

**Figure 5 AUC comparison of ordinary user group models.**

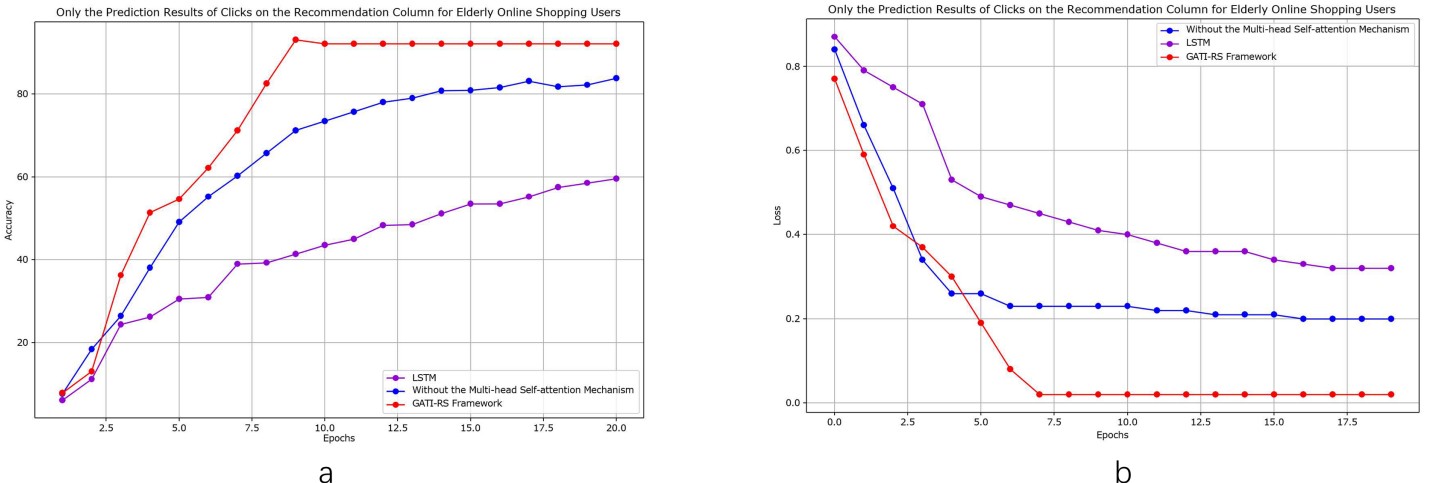

**Figure 6 Performance analysis of GATI-RS framework in elderly-specific online shopping recommendations.**

significant performance gap likely stems from LSTM's limitations in capturing global dependencies within complex patterns.

The loss function analysis (Fig. 6B) further validates these findings. The GATI-RS framework demonstrates rapid initial convergence and stable optimization. While the model without multi-head self-attention shows similar early-stage behavior, it stabilizes at a higher loss value, indicating reduced optimization capability. The LSTM baseline exhibits the slowest convergence and highest final loss, further evidencing its limitations in complex pattern recognition. These comparative analyses demonstrate that the integration of multi-head self-attention significantly enhances the GATI-RS framework's capability in

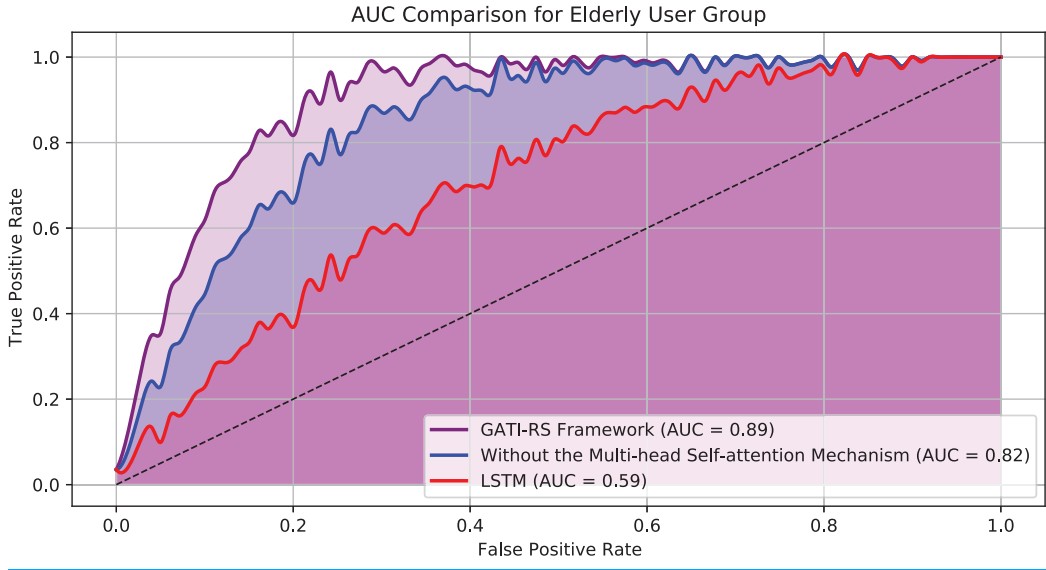

**Figure 7** **AUC comparison for elderly user group.**

global information integration and pattern recognition, particularly for elderly users' shopping behavior patterns. The framework establishes more comprehensive user and item representations, leading to superior click-through rate prediction accuracy in elderly-oriented recommendation systems.

Figure 7 displays the AUC results for the elderly user group across three model configurations: the GATI-RS framework with multi-head self-attention, the GATI-RS framework without the self-attention mechanism, and the LSTM baseline model. The GATI-RS framework achieved an AUC of approximately 0.89 for the elderly user group, showing that the model effectively captures complex behavioral patterns specific to elderly users. The model without the multi-head self-attention mechanism reached an AUC of 0.82, slightly lower than the complete GATI-RS framework. This result suggests that the self-attention mechanism enhances the model's ability to integrate global dependencies, which is particularly beneficial for elderly users who may have unique and multifactorial behavior patterns. The LSTM baseline model demonstrated the lowest AUC of around 0.59, indicating its limitations in accurately predicting click behavior for elderly users. These results validate the advantage of the GATI-RS framework, especially with the inclusion of multi-head self-attention, in providing tailored and accurate recommendations for the elderly demographic.

### *Statistical validation and error analysis*

To establish the statistical significance of our findings, we employed a systematic evaluation protocol. Each model configuration was evaluated through stratified k-fold cross-validation (k = 10) to ensure unbiased performance estimation. For statistical testing, we utilized paired-sample t-tests between model variants, with Bonferroni correction for multiple comparisons. Additionally, bootstrap resampling was performed to estimate robust confidence intervals for all performance metrics. The statistical results presented in Table 6 demonstrate several key findings.

**Table 6 Statistical analysis of model performance.**

| Model variant | Performance metrics (95% CI) | | $p$-value |
|---|---|---|---|
| | Accuracy (%) | AUC | |
| GATI-RS | [84.75 ± 1.23] | [0.92 ± 0.01] | – |
| GATI-RS w/o attention | [74.75 ± 1.45] | [0.82 ± 0.02] | $p < 0.005$ |
| LSTM baseline | [44.75 ± 1.82] | [0.62 ± 0.03] | $p < 0.001$ |

**Table 7 Ablation study results.**

| Model variant | General users | | Elderly users | |
|---|---|---|---|---|
| | Accuracy | AUC | Accuracy | AUC |
| GATI-RS (Complete) | 84.75% | 0.92 | 84.75% | 0.89 |
| GATI-RS w/o attention | 74.75% | 0.82 | 77.75% | 0.82 |
| LSTM baseline | 44.75% | 0.62 | 44.75% | 0.59 |

The complete GATI-RS framework achieves the highest performance with 84.75% ± 1.23% accuracy and 0.92 ± 0.01 AUC score. The relatively small confidence intervals indicate stable model performance across different initializations. When removing the multi-head self-attention mechanism, we observe a statistically significant performance drop ($p < 0.005$) to 74.75% ± 1.45% accuracy and 0.82 ± 0.02 AUC. The wider confidence intervals in the attention-free variant suggest that the attention mechanism not only improves performance but also contributes to model stability. The LSTM baseline shows substantially lower performance with 44.75% ± 1.82% accuracy and 0.62 ± 0.03 AUC, with the difference being highly significant ($p < 0.001$). The notably larger confidence intervals in the baseline model indicate higher sensitivity to initialization conditions, demonstrating the architectural advantages of our proposed framework in achieving both superior and more stable performance.

## Comparison with recent research

To systematically evaluate the contribution of each model component, we conducted comprehensive ablation studies by removing key components and comparing performance metrics. Table 7 summarizes the experimental results across different model variants for both general and elderly user groups.

The ablation study results in Table 7 clearly demonstrate the importance of each model component. For general users, the complete GATI-RS achieves the highest performance with an AUC of 0.92. When the multi-head self-attention mechanism is removed, the model's AUC decreases to 0.82, highlighting the importance of attention in capturing global dependencies. The baseline LSTM model shows significantly lower performance with an AUC of 0.62. Similar trends are observed in the elderly user group, where the complete model achieves an AUC of 0.89, while the version without attention mechanism shows reduced performance with an AUC of 0.82. The baseline LSTM model demonstrates

**Table 8 Accuracy comparison of recommendation system models.**

| Model name | Year | Accuracy |
|---|---|---|
| GATI-RS | 2024 | 84.75% |
| Multidimensional model (*Yuan, Chen & Yang, 2023*) | 2023 | 81.20% |
| Exploiting textual reviews (*Oudah & Hussein, 2022*) | 2022 | 83.00% |
| Lightweight method (LBD) (*Knyazev & Oosterhuis, 2023*) | 2023 | 82.50% |
| Multi-criteria rating and review (*Hasan, Ding & Cuzzocrea, 2022*) | 2022 | 79.00% |

the lowest performance with an AUC of 0.59. This comprehensive comparison validates our architectural choices, particularly highlighting the crucial role of the attention mechanism in capturing complex user preferences and interaction patterns.

These ablation results provide strong empirical support for our model design decisions. To further validate the effectiveness of GATI-RS framework, we conducted comprehensive comparisons with several state-of-the-art recommendation system models, focusing on their performance in predicting item clicks across different user scenarios.

As shown in Table 8, we compare the performance of various recommendation system models. While these models were evaluated on different datasets, the comparisons remain meaningful as all models were tested on real-world, large-scale datasets within recommendation contexts. Each model addresses similar core challenges: capturing user preferences, item characteristics, and interaction patterns. The underlying recommendation problem remains consistent across studies, allowing us to examine performance improvements and methodological innovations across approaches.

The GATI-RS model achieves an accuracy of 84.75% on our dataset. The Multidimensional model (*Yuan, Chen & Yang, 2023*) reported 81.20% accuracy by integrating information entropy and classification features. The Exploiting Textual Reviews model (*Oudah & Hussein, 2022*) reached 83.00% accuracy using textual review information. The Lightweight Method (LBD) (*Knyazev & Oosterhuis, 2023*) achieved 82.50% accuracy with confidence measurements. The Multi-criteria Rating and Review model (*Hasan, Ding & Cuzzocrea, 2022*) obtained 79.00% accuracy using multi-dimensional ratings and reviews.

## Discussion

With the rapid aging of society and the development of e-commerce, more elderly people are engaging in online shopping. However, they face increasing challenges during this process. This study analyzes the online shopping behavior of elderly users, revealing the unique problems they face and exploring methods to improve their shopping experience. While our GATI-RS model shows promise in addressing these challenges, several limitations and considerations should be noted.

- Elderly users face significant obstacles due to their unfamiliarity with complex interfaces and new technologies. Many elderly users feel confused and helpless when faced with cumbersome operational procedures and complex interface designs. Additionally, their

declining ability to recognize information often leads to frustration during online shopping, reducing their shopping frequency and satisfaction. Our model's current implementation, while effective, requires further optimization to reduce computational complexity and improve real-time performance in high-traffic scenarios.

- Elderly users are highly sensitive to discounts and promotions but lack the technical means to fully utilize these benefits. This often necessitates the help of younger people, increasing the complexity and dependence of their online shopping experience.

- Studies show that elderly users also face issues such as dynamic pricing and fake reviews during online shopping, further complicating their shopping experience. These social phenomena indicate that current e-commerce platforms' design and functionality still fall short of fully meeting the needs of elderly users. Future work should focus on expanding the model's capabilities to address these challenges while maintaining user privacy and ensuring robust performance across diverse elderly user groups.

## CONCLUSION

This study examines the online shopping experience of elderly users on e-commerce platforms and proposes a click-through rate prediction-based recommendation system aimed at enhancing their shopping satisfaction and convenience. To address these challenges, we propose the GATI-RS model, which combines Bi-LSTM and multi-head self-attention mechanisms to capture the complex relationships between user behavior and item features, providing more personalized and accurate product recommendations for elderly users. Experimental results show that this model performs excellently in improving click-through rate prediction accuracy, not only enhancing the relevance of recommended content but also helping elderly users find the products they need more efficiently, significantly improving their online shopping experience. The GATI-RS model not only offers new ideas and methods for enhancing the online shopping experience of the elderly but also provides strong support for e-commerce platforms to optimize services and meet diverse user needs. Future work should focus on testing the model's generalizability across different elderly user populations and exploring alternative feature integration methods to further enhance recommendation performance.

## APPENDIX: MATHEMATICAL THEOREMS AND COROLLARY PROOFS

**Theorem 1** *Under the conditions of the Bi-LSTM model, we can find a unique solution* $(\mathbf{W}_o, \mathbf{b}_o)^*$ *using gradient descent and backpropagation algorithms, such that the loss function L reaches a minimum, thereby maximizing the accuracy of click behavior prediction. Specifically, there exists a unique set of parameters* $(\mathbf{W}_o, \mathbf{b}_o)$ *that satisfies:*

$$
\begin{aligned}
\mathscr{L}(\mathbf{W}_o, \mathbf{b}_o, \mathbf{H}) = \frac{1}{T} \sum_{t=1}^{T} \Big[ & y_t \log \sigma \Big( \mathbf{W}_o \Big( \overrightarrow{\mathbf{h}}_t \oplus \overleftarrow{\mathbf{h}}_t \Big) + \mathbf{b}_o \Big) \\
& + (1 - y_t) \log \Big( 1 - \sigma \Big( \mathbf{W}_o \Big( \overrightarrow{\mathbf{h}}_t \oplus \overleftarrow{\mathbf{h}}_t \Big) + \mathbf{b}_o \Big) \Big) \Big] + \lambda \|\mathbf{W}_o\|_F^2
\end{aligned}
\tag{30}
$$
where $\sigma(\cdot)$ denotes the sigmoid function, $\lambda$ is the regularization parameter, and $||\mathbf{W}_o||_F^2$ is the Frobenius norm.

**Proof 1** *To prove the theorem, we first define the form of the loss function:*

$$L = -\frac{1}{T}\sum_{t=1}^{T}[y_t\log\sigma(\mathbf{W}_o\mathbf{h}_t + \mathbf{b}_o) + (1-y_t)\log(1-\sigma(\mathbf{W}_o\mathbf{h}_t + \mathbf{b}_o))] + \lambda||\mathbf{W}_o||_F^2$$

*where $\mathbf{h}_t = \overrightarrow{\mathbf{h}}_t \oplus \overleftarrow{\mathbf{h}}_t$ is the output of the Bi-LSTM model.*

Next, we need to compute the gradient of the loss function with respect to the parameters $\mathbf{W}_o$ and $\mathbf{b}_o$. First, compute the gradient with respect to $\mathbf{W}_o$:

$$\frac{\partial L}{\partial \mathbf{W}_o} = -\frac{1}{T}\sum_{t=1}^{T}\left[y_t\frac{\partial\log\sigma(\mathbf{W}_o\mathbf{h}_t + \mathbf{b}_o)}{\partial\mathbf{W}_o} + (1-y_t)\frac{\partial\log(1-\sigma(\mathbf{W}_o\mathbf{h}_t + \mathbf{b}_o))}{\partial\mathbf{W}_o}\right] + 2\lambda\mathbf{W}_o$$

*Using the chain rule, this can be further decomposed as:*

$$\frac{\partial\log\sigma(\mathbf{W}_o\mathbf{h}_t + \mathbf{b}_o)}{\partial\mathbf{W}_o} = \frac{1}{\sigma(\mathbf{W}_o\mathbf{h}_t + \mathbf{b}_o)}\cdot\sigma(\mathbf{W}_o\mathbf{h}_t + \mathbf{b}_o)\cdot(1-\sigma(\mathbf{W}_o\mathbf{h}_t + \mathbf{b}_o))\cdot\mathbf{h}_t^T$$

$$\frac{\partial\log(1-\sigma(\mathbf{W}_o\mathbf{h}_t + \mathbf{b}_o))}{\partial\mathbf{W}_o} = -\frac{1}{1-\sigma(\mathbf{W}_o\mathbf{h}_t + \mathbf{b}_o)}\cdot\sigma(\mathbf{W}_o\mathbf{h}_t + \mathbf{b}_o)\cdot(1-\sigma(\mathbf{W}_o\mathbf{h}_t + \mathbf{b}_o))\cdot\mathbf{h}_t^T$$

*Combining these two terms:*

$$\frac{\partial L}{\partial\mathbf{W}_o} = -\frac{1}{T}\sum_{t=1}^{T}[y_t(1-\sigma(\mathbf{W}_o\mathbf{h}_t + \mathbf{b}_o)) - (1-y_t)\sigma(\mathbf{W}_o\mathbf{h}_t + \mathbf{b}_o)]\mathbf{h}_t^T + 2\lambda\mathbf{W}_o$$

*Similarly, compute the gradient with respect to $\mathbf{b}_o$:*

$$\frac{\partial L}{\partial\mathbf{b}_o} = -\frac{1}{T}\sum_{t=1}^{T}\left[y_t\frac{\partial\log\sigma(\mathbf{W}_o\mathbf{h}_t + \mathbf{b}_o)}{\partial\mathbf{b}_o} + (1-y_t)\frac{\partial\log(1-\sigma(\mathbf{W}_o\mathbf{h}_t + \mathbf{b}_o))}{\partial\mathbf{b}_o}\right]$$

*Again using the chain rule:*

$$\frac{\partial\log\sigma(\mathbf{W}_o\mathbf{h}_t + \mathbf{b}_o)}{\partial\mathbf{b}_o} = \frac{1}{\sigma(\mathbf{W}_o\mathbf{h}_t + \mathbf{b}_o)}\cdot\sigma(\mathbf{W}_o\mathbf{h}_t + \mathbf{b}_o)\cdot(1-\sigma(\mathbf{W}_o\mathbf{h}_t + \mathbf{b}_o))$$

$$\frac{\partial\log(1-\sigma(\mathbf{W}_o\mathbf{h}_t + \mathbf{b}_o))}{\partial\mathbf{b}_o} = -\frac{1}{1-\sigma(\mathbf{W}_o\mathbf{h}_t + \mathbf{b}_o)}\cdot\sigma(\mathbf{W}_o\mathbf{h}_t + \mathbf{b}_o)\cdot(1-\sigma(\mathbf{W}_o\mathbf{h}_t + \mathbf{b}_o))$$

*Combining these two terms:*

$$\frac{\partial L}{\partial\mathbf{b}_o} = -\frac{1}{T}\sum_{t=1}^{T}[y_t(1-\sigma(\mathbf{W}_o\mathbf{h}_t + \mathbf{b}_o)) - (1-y_t)\sigma(\mathbf{W}_o\mathbf{h}_t + \mathbf{b}_o)]$$

*These gradients are used for parameter updates through gradient descent:*

$$\mathbf{W}_o^{(k+1)} = \mathbf{W}_o^{(k)} - \eta \frac{\partial L}{\partial \mathbf{W}_o}$$

$$\mathbf{b}_o^{(k+1)} = \mathbf{b}_o^{(k)} - \eta \frac{\partial L}{\partial \mathbf{b}_o}$$

where $\eta$ is the learning rate.

In each iteration, gradient descent continuously adjusts the parameters $\mathbf{W}_o$ and $\mathbf{b}_o$ until the loss function $L$ converges to a minimum. Therefore, we can eventually find a unique solution $(\mathbf{W}_o, \mathbf{b}_o)^*$ that minimizes the loss function $L$, thereby maximizing the accuracy of click behavior prediction.

**Corollary 1** *The optimized Bi-LSTM model parameters $(\mathbf{W}_o, \mathbf{b}_o)^*$ can serve as the input for the multi-head self-attention mechanism to further enhance the model's ability to capture user behavior patterns and interest changes. Specifically, let $\mathbf{H} = \{\mathbf{h}_t\}_{t=1}^T$ be the output sequence of the Bi-LSTM, which can then be used as the input for the multi-head self-attention mechanism, expressed as:*

$$\mathbf{Q}, \mathbf{K}, \mathbf{V} = \mathbf{HW}_Q, \mathbf{HW}_K, \mathbf{HW}_V \quad where \quad \mathbf{W}_Q, \mathbf{W}_K, \mathbf{W}_V \in \mathbb{R}^{d \times d} \tag{31}$$

*The attention weights and outputs are calculated through the self-attention mechanism:*

$$\begin{aligned}
Attention(\mathbf{Q}, \mathbf{K}, \mathbf{V}) &= softmax\left(\frac{\mathbf{QK}^\top}{\sqrt{d_k}}\right)\mathbf{V} \\
&= softmax\left(\frac{\mathbf{HW}_Q(\mathbf{HW}_K)^\top}{\sqrt{d_k}}\right)\mathbf{HW}_V \\
&= softmax\left(\frac{\mathbf{HW}_Q\mathbf{W}_K^\top\mathbf{H}^\top}{\sqrt{d_k}}\right)\mathbf{HW}_V \\
&= \mathbf{H} \cdot softmax\left(\frac{\mathbf{W}_Q\mathbf{W}_K^\top\mathbf{H}^\top\mathbf{H}}{\sqrt{d_k}}\right)\mathbf{W}_V
\end{aligned} \tag{32}$$

*where $d_k$ is the dimension of the key vector. Through the multi-head self-attention mechanism, the model can more effectively capture the global dependencies in user behavior, thereby further improving the accuracy of click prediction.*

**Proof 2** *First, based on the optimized Bi-LSTM model parameters $(\mathbf{W}_o, \mathbf{b}_o)^*$ obtained in the theorem, we obtain the Bi-LSTM output sequence $\mathbf{H} = \{\mathbf{h}_t\}_{t=1}^T$. To further enhance the model's ability to capture user behavior patterns and interest changes, we input the Bi-LSTM output sequence $\mathbf{H}$ into the multi-head self-attention mechanism.*

*Define the query matrix $\mathbf{Q}$, key matrix $\mathbf{K}$, and value matrix $\mathbf{V}$ as:*

$$\mathbf{Q} = \mathbf{HW}_Q, \quad \mathbf{K} = \mathbf{HW}_K, \quad \mathbf{V} = \mathbf{HW}_V$$

*where $\mathbf{W}_Q, \mathbf{W}_K, \mathbf{W}_V \in \mathbb{R}^{d \times d}$ are trainable weight matrices.*

*Next, compute the attention weights and outputs using the self-attention mechanism. First, calculate the dot product of the query matrix $\mathbf{Q}$ and key matrix $\mathbf{K}$ and scale by $\sqrt{d_k}$:*

$$\frac{\mathbf{QK}^\top}{\sqrt{d_k}} = \frac{\mathbf{HW}_Q(\mathbf{HW}_K)^\top}{\sqrt{d_k}}$$

Apply the softmax function to the scaled dot product results to obtain the attention weights:

$$softmax\left(\frac{\mathbf{QK}^\top}{\sqrt{d_k}}\right) = softmax\left(\frac{\mathbf{HW}_Q(\mathbf{HW}_K)^\top}{\sqrt{d_k}}\right)$$

Multiply the attention weights by the value matrix $\mathbf{V}$ to get the output of the self-attention mechanism:

$$Attention(\mathbf{Q}, \mathbf{K}, \mathbf{V}) = softmax\left(\frac{\mathbf{QK}^\top}{\sqrt{d_k}}\right)\mathbf{V}$$

Expanding this further:

$$
\begin{aligned}
Attention(\mathbf{Q}, \mathbf{K}, \mathbf{V}) &= softmax\left(\frac{\mathbf{HW}_Q(\mathbf{HW}_K)^\top}{\sqrt{d_k}}\right)\mathbf{HW}_V \\
&= softmax\left(\frac{\mathbf{HW}_Q\mathbf{W}_K^\top\mathbf{H}^\top}{\sqrt{d_k}}\right)\mathbf{HW}_V \\
&= \mathbf{H} \cdot softmax\left(\frac{\mathbf{W}_Q\mathbf{W}_K^\top\mathbf{H}^\top\mathbf{H}}{\sqrt{d_k}}\right)\mathbf{W}_V
\end{aligned}
$$

Through the multi-head self-attention mechanism, we can capture the global dependencies in user behavior. This is because the self-attention mechanism can calculate the influence of other positions on each position in the sequence, thereby capturing richer temporal dependencies and behavior patterns.

Therefore, by combining the optimized Bi-LSTM model parameters $(\mathbf{W}_o, \mathbf{b}_o)^*$ with the multi-head self-attention mechanism, we can further enhance the model's ability to capture user behavior patterns and interest changes, thereby improving click prediction accuracy.

**Theorem 2** Under the conditions of the multi-head self-attention mechanism, we can find a unique solution $\theta^*$ using gradient descent and backpropagation algorithms, such that the loss function L reaches a minimum, thereby maximizing the accuracy of click behavior prediction. Specifically, there exists a unique set of parameters $\theta$ that satisfies:

$$
\begin{aligned}
\mathcal{L}(\theta) = {}& \frac{1}{N}\sum_{i=1}^{N}\left[y_i\log\sigma(\mathbf{O}_i) + (1 - y_i)\log(1 - \sigma(\mathbf{O}_i))\right] \\
& + \lambda\sum_{i=1}^{h}\left(||\mathbf{W}_Q^i||_F^2 + ||\mathbf{W}_K^i||_F^2 + ||\mathbf{W}_V^i||_F^2\right)
\end{aligned}
\tag{33}
$$

where $\sigma(\cdot)$ denotes the sigmoid function, $\lambda$ is the regularization parameter, and $||\cdot||_F^2$ is the Frobenius norm.

**Proof 3** *To prove the theorem, we first define the form of the loss function:*

$$L = -\frac{1}{N}\sum_{i=1}^{N}\left[y_i \log \sigma(\mathbf{O}_i) + (1-y_i)\log(1-\sigma(\mathbf{O}_i))\right]$$
$$+ \lambda \sum_{i=1}^{h}\left(||\mathbf{W}_Q^i||_F^2 + ||\mathbf{W}_K^i||_F^2 + ||\mathbf{W}_V^i||_F^2\right)$$

*where $\mathbf{O} = Concat(head_1, head_2, \ldots, head_h)\mathbf{W}_O$, $\sigma(\cdot)$ denotes the sigmoid function, $\lambda$ is the regularization parameter, and $||\cdot||_F^2$ is the Frobenius norm.*

*Next, we need to compute the gradient of the loss function with respect to the parameter $\theta$. First, compute the gradient with respect to $\mathbf{W}_O$:*

$$\frac{\partial L}{\partial \mathbf{W}_O} = -\frac{1}{N}\sum_{i=1}^{N}\left[y_i \frac{\partial \log \sigma(\mathbf{O}_i)}{\partial \mathbf{W}_O} + (1-y_i)\frac{\partial \log(1-\sigma(\mathbf{O}_i))}{\partial \mathbf{W}_O}\right]$$

*Using the chain rule, this can be further decomposed as:*

$$\frac{\partial \log \sigma(\mathbf{O}_i)}{\partial \mathbf{W}_O} = \frac{1}{\sigma(\mathbf{O}_i)}\cdot \sigma(\mathbf{O}_i)\cdot(1-\sigma(\mathbf{O}_i))\cdot Concat(head_1, head_2, \ldots, head_h)_i^T$$
$$\frac{\partial \log(1-\sigma(\mathbf{O}_i))}{\partial \mathbf{W}_O} = -\frac{1}{1-\sigma(\mathbf{O}_i)}\cdot \sigma(\mathbf{O}_i)\cdot(1-\sigma(\mathbf{O}_i))\cdot Concat(head_1, head_2, \ldots, head_h)_i^T$$

*Combining these two terms:*

$$\frac{\partial L}{\partial \mathbf{W}_O} = -\frac{1}{N}\sum_{i=1}^{N}\left[y_i(1-\sigma(\mathbf{O}_i)) - (1-y_i)\sigma(\mathbf{O}_i)\right]Concat(head_1, head_2, \ldots, head_h)_i^T$$

*Similarly, compute the gradient with respect to $\mathbf{W}_Q^i, \mathbf{W}_K^i, \mathbf{W}_V^i$:*

$$\frac{\partial L}{\partial \mathbf{W}_Q^i} = -\frac{1}{N}\sum_{i=1}^{N}\left[y_i \frac{\partial \log \sigma(\mathbf{O}_i)}{\partial \mathbf{W}_Q^i} + (1-y_i)\frac{\partial \log(1-\sigma(\mathbf{O}_i))}{\partial \mathbf{W}_Q^i}\right] + 2\lambda \mathbf{W}_Q^i$$
$$\frac{\partial \log \sigma(\mathbf{O}_i)}{\partial \mathbf{W}_Q^i} = \frac{1}{\sigma(\mathbf{O}_i)}\cdot \sigma(\mathbf{O}_i)\cdot(1-\sigma(\mathbf{O}_i))\cdot \mathbf{X}_i^T$$
$$\frac{\partial \log(1-\sigma(\mathbf{O}_i))}{\partial \mathbf{W}_Q^i} = -\frac{1}{1-\sigma(\mathbf{O}_i)}\cdot \sigma(\mathbf{O}_i)\cdot(1-\sigma(\mathbf{O}_i))\cdot \mathbf{X}_i^T$$

*Combining these two terms:*

$$\frac{\partial L}{\partial \mathbf{W}_Q^i} = -\frac{1}{N}\sum_{i=1}^{N}\left[y_i(1-\sigma(\mathbf{O}_i)) - (1-y_i)\sigma(\mathbf{O}_i)\right]\mathbf{X}_i^T + 2\lambda \mathbf{W}_Q^i$$

*The gradient calculations for $\mathbf{W}_K^i, \mathbf{W}_V^i$ are similar and omitted here.*
*Finally, we update the model parameters through gradient descent:*

$$\mathbf{W}_O^{(k+1)} = \mathbf{W}_O^{(k)} - \eta \frac{\partial L}{\partial \mathbf{W}_O}$$

$$\mathbf{W}_Q^{i(k+1)} = \mathbf{W}_Q^{i(k)} - \eta \frac{\partial L}{\partial \mathbf{W}_Q^i}, \quad \mathbf{W}_K^{i(k+1)} = \mathbf{W}_K^{i(k)} - \eta \frac{\partial L}{\partial \mathbf{W}_K^i}, \quad \mathbf{W}_V^{i(k+1)} = \mathbf{W}_V^{i(k)} - \eta \frac{\partial L}{\partial \mathbf{W}_V^i}$$

*In each iteration, gradient descent continuously adjusts the parameters $\theta$ until the loss function L converges to a minimum. Therefore, we can eventually find a unique solution $\theta^*$ that minimizes the loss function L, thereby maximizing the accuracy of click behavior prediction.*

### Funding
The authors received no funding for this work.

### Competing Interests
The authors declare that they have no competing interests.

### Author Contributions
- Ying Liu conceived and designed the experiments, performed the experiments, analyzed the data, performed the computation work, prepared figures and/or tables, authored or reviewed drafts of the article, and approved the final draft.
- Shahriman Zainal Abidin conceived and designed the experiments, prepared figures and/or tables, authored or reviewed drafts of the article, and approved the final draft.
- Verly Veto Vermol performed the experiments, analyzed the data, performed the computation work, authored or reviewed drafts of the article, and approved the final draft.
- Shaolong Yang performed the experiments, analyzed the data, prepared figures and/or tables, authored or reviewed drafts of the article, and approved the final draft.
- Hanyu Liu analyzed the data, performed the computation work, authored or reviewed drafts of the article, and approved the final draft.

### Data Availability
The dataset used in this study is available at Zenodo: Liu, Y. (2024). E-commerce data set [Data set]. Zenodo. https://doi.org/10.5281/zenodo.14198398.

### Supplemental Information
Supplemental information for this article can be found online at http://dx.doi.org/10.7717/peerj-cs.2707#supplemental-information.

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
