# Peer review of "GATI-RS model using Bi-LSTM and multi-head attention mechanism to enhance online shopping experience for the elderly with accurate click-through rate prediction"

_PeerJ Computer Science, doi:10.7717/peerj-cs.2707_

## Round 0.1 · original submission · Major Revisions

Dear authors,

Thank you for the submission. The reviewers’ comments are now available. It is not suggested that your article be published in its current format. We do, however, advise you to revise the paper in light of the reviewers’ comments and concerns before resubmitting it. The followings should also be addressed:

1. Please pay special attention to the usage of abbreviations. Spell out the full term at its first mention, indicate its abbreviation in parenthesis and use the abbreviation from then on.
2. Equations should be used with correct equation number. Many of the equations are part of the related sentences. Attention is needed for correct sentence formation.
3. Advantages and disadvantages of the methods should be clarified. Practical advantages should be indicated and the limitations of the research should be discussed.
4. In the conclusions, please state explicitly what lessons can be learnt from this study and then describe in more detail the future research directions.

Best wishes,

Reviewer 1 ·

Basic reporting

The manuscript addresses the critical issue of enhancing the online shopping experience for the elderly using a recommendation system based on click-through rate (CTR) prediction. While the paper explores an intriguing application of deep learning and attention mechanisms, it exhibits numerous deficiencies, inconsistencies, and weaknesses that need to be resolved for the manuscript to meet the standard of a high-impact journal publication.
The introduction provides a general overview of the challenges faced by elderly online shoppers and attempts to justify the need for a tailored recommendation system. However, the motivation for selecting the Bi-LSTM and multi-head self-attention mechanism over other models is inadequately explained. Although the authors mention that these mechanisms can handle global feature dependencies, they do not delve into how these aspects are particularly crucial for elderly users. Furthermore, the introduction lacks a clear articulation of the knowledge gap the proposed study aims to fill. While the challenges of online shopping for the elderly are briefly described, there is little to no critical engagement with the existing literature on recommendation systems. The authors should offer a more specific problem statement and a stronger justification for why their chosen model architecture is well-suited to address the identified issues.
The literature review is notably superficial and fails to critically engage with prior works in the field. Although the authors reference various studies involving machine learning for recommendation systems, there is a lack of depth in the discussion. The review lists studies without adequately analyzing their methodologies, strengths, and limitations, especially concerning the elderly demographic in e-commerce. Furthermore, the authors do not provide a comparison of alternative methods to handle temporal dependencies or global feature integration in click-through rate prediction. The review also lacks discussion on the limitations of existing models, such as computational complexity or interpretability, and how the proposed GATI-RS model addresses these gaps. For a comprehensive literature review, the authors must include a critical analysis of related work and clearly identify how their approach advances the state-of-the-art in recommendation systems tailored for elderly users.

Experimental design

The manuscript mentions the use of a custom e-commerce dataset, but it provides limited details regarding data acquisition, structure, and quality. Important aspects such as the sources of user behavior data, the granularity of the item features, and the criteria used for data collection are either missing or insufficiently described. Although the authors indicate that standardization and one-hot encoding were employed during data preprocessing, the rationale behind these specific choices and their impact on the final model performance are not discussed. Additionally, the treatment of missing data through mean imputation raises concerns, as this approach may introduce bias or obscure significant patterns in user behavior. The authors should elaborate on the data preprocessing steps, including why certain methods were chosen and how these steps address potential inconsistencies or noise in the dataset.
The description of the GATI-RS model is fragmented and lacks critical technical details. The authors claim to use a combination of Bi-LSTM and multi-head self-attention mechanisms but fail to provide a clear explanation of how these components are integrated into the overall architecture. For example, there is no information about the number of layers, hidden units, or attention heads, nor is there a detailed diagram illustrating the model's internal workings. Furthermore, the paper does not discuss the potential limitations of the model architecture, such as the computational complexity introduced by the multi-head self-attention mechanism, or how it scales with an increasing dataset size. The lack of clarity in the model's design hinders reproducibility and leaves questions regarding its novelty and effectiveness.
The manuscript provides only a cursory description of the training process. Key hyperparameters, such as the learning rate, batch size, and regularization methods, are mentioned without any justification for their selection. Additionally, while the authors state that early stopping and cross-entropy loss were used, they do not explain the criteria for early stopping or how the model was evaluated during training. The absence of detailed explanations of hyperparameter tuning or ablation studies to test the model's sensitivity to these parameters is a major shortcoming. Moreover, there is no discussion on strategies to mitigate overfitting, which is a critical concern given the size of the dataset and the complexity of the model.
The experimental design is poorly structured, and the choice of baseline models lacks depth. The manuscript states that the authors used LSTM as a baseline, but there is no discussion on why this specific model was chosen or whether other potential baselines, such as transformer-based models, were considered. The lack of comparative analysis against more sophisticated baseline models raises questions about the robustness of the reported results. Furthermore, the paper does not address how the authors handled data partitioning into training, validation, and test sets to ensure unbiased evaluation, nor does it discuss the evaluation criteria used during cross-validation.

Validity of the findings

While the authors report accuracy and loss as performance metrics, they fail to provide a more nuanced analysis of model performance using other metrics such as precision, recall, or F1-score, which are particularly relevant in click-through rate prediction. The emphasis on accuracy alone is problematic, as it does not capture the true performance of the recommendation model in handling diverse user behaviors and preferences. The authors do not provide statistical validation (e.g., confidence intervals, p-values) to support their claims of improvement over the baseline models, nor do they include an error analysis to identify where the model performs poorly. This omission is critical, as it leaves readers unable to assess the model's generalizability and potential limitations.
The discussion section is overly optimistic and lacks a critical examination of the study's limitations. The authors make broad claims about the superiority of the GATI-RS model but do not address fundamental questions about the model's real-world applicability, computational requirements, or robustness to variations in user behavior data. Furthermore, the discussion fails to engage with the practical challenges of deploying such a system in an operational e-commerce setting, such as data privacy, the need for model interpretability, and the potential for bias in recommendations. The authors should offer a more balanced discussion, acknowledging the constraints of their model and suggesting how future work could address these issues.
Moreover, the conclusion does not highlight the limitations of the study, such as the narrow focus on a custom dataset and the lack of external validation. A more reflective conclusion that outlines future research directions, such as exploring other feature integration methods or testing the model on a diverse set of e-commerce datasets, is needed.

Additional comments

Overall, this manuscript has potential but requires significant revisions. The primary concerns include the lack of a comprehensive literature review, incomplete methodological details, insufficient statistical validation of results, and an overly optimistic discussion that does not critically assess the model's limitations. The authors should consider the following specific suggestions for improvement:

• Provide a stronger motivation for using Bi-LSTM and multi-head attention mechanisms and clearly define the problem being addressed.
• Conduct a critical analysis of related work and clearly outline how the proposed method differs and advances the field.
• Provide a detailed description of the model architecture, training process, hyperparameter tuning, and strategies to prevent overfitting.
• Include a comparison with more sophisticated baseline models and provide a comprehensive evaluation using multiple performance metrics.
• Report statistical measures such as confidence intervals to support claims of performance improvement.
• Critically discuss the model's limitations and suggest practical ways to overcome them in future research.
Substantial revisions are necessary for this manuscript to make a meaningful contribution to the field of e-commerce recommendation systems tailored to elderly users. Without these changes, the study's impact and utility remain limited and speculative.

Reviewer 2 ·

Basic reporting

The introduction provides a solid motivation for the study; however, the main contribution and novelty of the work remain unclear. For instance, is the author suggesting that other CTR prediction models do not utilize age-related features? If that is the case, this claim should be substantiated by citing relevant research articles that demonstrate the gap in existing literature.

The main objective of the article is articulated in the following sentence:
'This study proposes a recommendation system (GATI-RS) combining Bi-LSTM and a multi-head self-attention mechanism, focusing on the unique needs of elderly users. The multi-head self-attention mechanism excels in handling global feature dependencies, providing more comprehensive feature integration, and enabling the model to better serve elderly users.'

While this objective is clear, the novelty of the proposed approach should be further clarified, specifically how the combination of Bi-LSTM and multi-head self-attention offers a distinct advantage over existing models, particularly for elderly users, and how this is measured. Additional details on how the model better addresses the needs of elderly users compared to traditional methods would strengthen the paper’s contribution. Also, a clearer definition of how you will be achieving this improvement is needed.

Experimental design

The author needs to demonstrate the improvement in the model when integrating age as a feature by either experimenting with features that include age groups and excluding age groups or by training the model separately for each age group. This should also be applied to the base models for a fair comparison. Only then can we conclude whether there is a clear advantage to incorporating this feature, particularly for elderly individuals.

In terms of evaluation, relying solely on accuracy as the performance metric is insufficient, especially if this is framed as a classification problem. As seen in similar research, such as the DeepFM paper (Guo et al., 2017), AUC (Area Under the ROC Curve) is often used alongside accuracy to provide a more comprehensive evaluation of model performance. AUC is particularly important for classification tasks, as it accounts for true positive and false positive rates across various thresholds, unlike accuracy, which focuses on a fixed threshold.

Validity of the findings

The experiments and evaluations in the paper have not been performed satisfactorily. To reconsider this article, the experimental design should be improved with the following adjustments:

Alignment with the paper's objective: How does the experimental design align with the paper’s stated objective? Currently, it appears that only the inclusion of age as a new feature is being tested. If that’s the case, where are the results for the model without the age feature? A direct comparison between these two scenarios is essential to justify the effectiveness of integrating age.

Evaluation metrics for CTR prediction: What other evaluation metrics can be utilized in assessing CTR prediction? Accuracy alone is not sufficient for evaluating model performance in this context. Metrics like AUC (Area Under the ROC Curve), would provide a more comprehensive understanding of the model’s strengths and weaknesses.

Improvement for elderly users vs. overall performance: How can we determine if the model’s outcome is specifically improved for elderly users, or if it is merely performing better overall across all age groups? More detailed age-group-specific evaluations are necessary to show whether the objective of improving the experience for elderly individuals has been met.

Reviewer 3 ·

Basic reporting

In the present paper the authors present a model that aims to excel in click-to-rate (CRT) prediction and indirectly enhances online shopping experience of the elderly. The idea and the approach are interesting and might be of interest for the audience of the journal. Unfortunately I can not suggest publication of the manuscript in its present form since, in my opinion, it does not meet the standards of an acknowledged scientific journal such as PeerJ CS.

My main objections are the presented in the following list.

P1. The manuscript, in its current form, appears overly lengthy and contains some outdated/obsolete content. As a result, in my opinion the reading experience is somewhat tedious and daunting, which may hinder the engagement of the audience. For example, the presenation of the theoretical part of the model, e.g. Bi-LSTM and Attention (Sec 2 and Appendix) should be limited only to the absolutely basic presentation of the models and to the adaptations made by the authors. Sec 2 should be developed around the algorithm presented in page 13. On the other hand the authors should made a more clear and in a concise presentation of the application and how it spports the online experience of the elderly. Also, thay should try to address the possible expainability of the models applied.

P2. In my opinion, figs 1 and 2 are rather obsolete. Users should focus in fig 3 to support their methodology (it is elder not old). Figs 4, 5, 6 and 7 must be redone so that the different series of data are clear for the reader, and I suggest being presented as a single figure with 4 subfigures.

P3. As far as the dataset is concerned, authors present just a link, there is no presentation of the dataset’s features and its size and there is no hint about its origin – it was made by the authors?

P4. Why didn’t they try Bi-LSTM without attention but simple LSTM? Also, they should present in more detail the (without) Multi-head self-attention mechanism.

P5. Authors should be careful about the comparison between different models for time and space complexity (Tab 2) and accuracy (Tab 5)

P6. In Tab 5 the accuracies presented are on the same dataset? It is not clear from the presentation and if not, the Tab has a fundamental flaw. Authors should be very careful on this and should make it absolutely clear in the manuscript.

P7. The manuscript would benefit from a thorough review to polish the use of English and correct typographical errors; they suggest a lack of the required attention to detail in the presentation.

Experimental design

Methods described with sufficient detail & information to replicate (code, dataset, computing infrastructure, reproduction script, etc.). / VERY GOOD

Is there a discussion on data preprocessing and is it sufficient/required? / VERY GOOD

Are the evaluation methods, assessment metrics, and model selection methods adequately described? / GOOD

Validity of the findings

Impact and novelty is not assessed. Meaningful replication is encouraged where rationale & benefit to the field is clearly stated. / GOOD

Are the experiments and evaluations performed satisfactorily? / GOOD

Does the Conclusion identify unresolved questions / limitations/ future directions? / GOOD

Comparison with relevant state-of-the-art works / WEAK

---

## Round 0.2 · Major Revisions

Dear Authors,

Your paper seems improved, however according to Reviewer 1, your paper still needs revision. Please make necessary additions and modifications and resubmit your paper.

Warm regards,

Reviewer 1 ·

Basic reporting

The authors have improved the paper. However, the authors have failed to discuss Usability and Security Testing of Online Links: A Framework for Click-Through Rate Prediction Using Deep Learning, which is a serious oversight. The authors have not adequately shown how their work advances the state-of-the-art or addresses specific gaps highlighted by the reviewers.

Experimental design

The authors reject suggestions for additional experiments, such as testing the impact of excluding age-related features or comparing Bi-LSTM without attention, on the grounds that current comparisons suffice. This weakens the claim of novelty and fails to adequately demonstrate the model's advantages.
To improve further, I suggest to include ablation studies or comparisons with other models to substantiate claims about the superiority of the Bi-LSTM and attention mechanisms.

Validity of the findings

The rebuttal does not address the broader concern about the lack of detailed error analysis or statistical validation of results (e.g., confidence intervals or significance tests).

Additional comments

Although figures were revised, the rebuttal does not clarify whether improvements were made to enhance their interpretability or alignment with the narrative.

---

## Round 0.3 · accepted · Accept

Dear Authors,

Thank you for clearly addressing the reviewers' comments. Your paper now seems ready for publication.

Best wishes,

Reviewer 1 ·

Basic reporting

The authors have revised well. I have no further comments.

Experimental design

The authors have revised well. I have no further comments.

Validity of the findings

The authors have revised well. I have no further comments.

Additional comments

I recommend to accept.

Reviewer 3 ·

Basic reporting

In the present paper the authors present a model that aims to excel in click-to-rate (CRT) prediction and indirectly enhances online shopping experience of the elderly. The idea and the approach are interesting and might be of interest for the audience of the journal. After two revision rounds the quality of the manuscript is substantially improved. I am satisfied with authors' responses to reviewers' comments; they have addressed most of the key concerns raised in the previous rounds. Therefore, I recommend the manuscript for publication in PeerJ CS.

Experimental design

Methods described with sufficient detail & information to replicate (code, dataset, computing infrastructure, reproduction script, etc.). / VERY GOOD

Is there a discussion on data preprocessing and is it sufficient/required? / VERY GOOD

Are the evaluation methods, assessment metrics, and model selection methods adequately described? / VERY GOOD

Validity of the findings

Impact and novelty is not assessed. Meaningful replication is encouraged where rationale & benefit to the field is clearly stated. / GOOD

Are the experiments and evaluations performed satisfactorily? / VERY GOOD

Does the Conclusion identify unresolved questions / limitations/ future directions? / VERY GOOD